# STORM: Synergistic Cross-Scale Spatio-Temporal Modeling for Weather Forecasting

**Qihe Huang**[1,2,*] **Zhengyang Zhou**[1,2,*] **Yangze Li**[1], **Jiaming Ma**[1], **Kuo Yang**[1], **Binwu Wang**[1,2],
**Xu Wang**[1,2], **Yang Wang**[1,2,*]

[1]University of Science and Technology of China (USTC), Hefei, Anhui, China
[2]Suzhou Institute for Advanced Research, USTC, Suzhou, Jiangsu, China
`hqh@mail.ustc.edu.cn`

## Abstract

Accurate weather forecasting is crucial for climate research, disaster mitigation, and societal planning. Despite recent progress with deep learning, global atmospheric data remain uniquely challenging since weather dynamics evolve across heterogeneous spatial and temporal scales ranging from planetary circulations to localized phenomena. Capturing such cross-scale interactions within a unified framework remains an open problem. To address this gap, we propose **STORM**, a synergistic cross-scale spatio-temporal model that disentangles atmospheric variations into multiple scales to uncover scale-specific dependencies. In addition, it enables coherent forecasting across multiple resolutions, maintaining consistent temporal evolution. Experiments on benchmark datasets demonstrate that STORM consistently delivers superior performance across both global and regional settings, as well as for short- and long-term forecasts. The code is available at https://github.com/h505023992/STORM.

## 1 Introduction

Accurate weather forecasting plays a pivotal role in climate research, disaster mitigation, and supporting decision-making in agriculture, energy, and public safety (Kuligowski & Barros, 1998; Baboo & Shereef, 2010; Lam et al., 2022; Gao et al., 2025). Traditional Numerical Weather Prediction (NWP) systems (Bauer et al., 2015) are grounded in the numerical solution of atmospheric dynamics (Buzzicotti et al., 2023), ensuring physical consistency across scales (Achatz et al., 2024). However, with the rapid growth of observational data and the demand for high-resolution, long-horizon forecasts, NWP approaches are increasingly constrained by high computational costs and difficulties in leveraging data-driven knowledge at scale (Rasp et al., 2023).

Deep learning (DL) has recently emerged as a promising paradigm for spatio-temporal forecasting (Yu et al., 2018; Wu et al., 2019; Guo et al., 2022; Wang et al., 2024a; Ma et al., 2025c;b;a;d;f;e; 2026a;b; Huang et al., 2025b;a;c; 2024a;c; Lin et al., 2023). Early efforts, such as ConvLSTM (Shi et al., 2015) and PredRNN (Wang et al., 2017), demonstrated the potential of neural architectures for regional precipitation prediction. More recent large-scale weather models, such as Pangu-Weather (Bi et al., 2023), GraphCast (Lam et al., 2022), and FourCastNet (Pathak et al., 2022), have achieved impressive advances in global forecasting and even in predicting extreme events (Chen et al., 2023b). Despite these successes, current DL-based approaches still face several critical limitations: ❶ **Multi-scale heterogeneity.** Global atmospheric circulation evolves at coarse scales, while regional variability emerges at finer scales. Balancing these dynamics within a single model remains challenging (Zhou et al., 2023). ❷ **Diverse temporal evolution.** Distinct scales exhibit different temporal dynamics, and existing models struggle to learn coherent cross-scale evolution, often leading to degraded performance in long-term forecasts (Wang et al., 2024b). ❸ **Weak cross-scale interaction.** Most methods process different spatial or temporal scales independently, overlooking their complementary roles and the synergistic effects across scales (Huang et al., 2024b). To address these challenges, we propose **STORM**, a synergistic cross-scale spatio-temporal method for accurate weather forecasting. As shown in Figure 1, STORM explicitly disentangles atmospheric variations across fine-to-coarse

---

*Zhengyang Zhou and Yang Wang are corresponding authors.

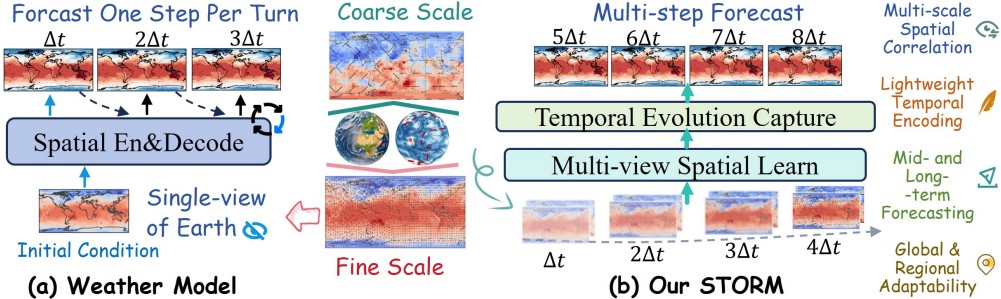

Figure 1: Comparison of main stream weather forecasting model and our STORM.

resolutions and integrates them via a cross-scale collaboration mechanism with temporal evolution learning. This design enables the model to capture global circulation patterns while preserving local high-frequency details, thereby enhancing both short- and long-term predictive skill. Through extensive experiments on benchmark meteorological datasets, we demonstrate that STORM achieves state-of-the-art forecasting performance. Our main contributions are summarized as follows:

- To address the challenge scale coupling in atmospheric data, we propose the first spatio-temporal modeling framework for weather forecasting. In particular, (i) it emphasizes cross-scale spatial learning to capture both unique local patterns and shared dependencies across different spatial resolutions, and (ii) it focuses on multi-step temporal evolution, rather than merely learning a single-step mapping from past to future states.

- We introduce STORM, a unified spatio-temporal framework for weather forecasting. It comprises (i) a *Hierarchical Earth Embedder* to build multi-resolution representations, (ii) a *Scale-Bridging Spatio-Temporal Encoder* to jointly model temporal evolution and spatial interactions across scales, and (iii) a *Level-Aligned Forecasting Decoder* to generate coherent multi-scale future predictions, enabling accurate decomposition, integration, and reconstruction of atmospheric dynamics.

- Extensive experiments on ERA5 datasets at multiple spatial resolutions ($5.625°$, $1°$, and $0.25°$) show that STORM consistently outperforms existing baselines across both global and regional domains. The framework demonstrates superior accuracy for short-term (hours) as well as long-term (several days) forecasts, effectively capturing fine-scale local patterns, medium-scale regional structures, and large-scale planetary circulations.

## 2 RELATED WORK

**Global Weather Forecasting** Global weather forecasting has seen rapid progress with the adoption of deep learning models Weyn et al. (2021); Nguyen et al. (2023). FourCastNet (Pathak et al., 2022), built upon Fourier neural operators, achieves forecasts comparable to numerical methods like IFS while being orders of magnitude faster . Pangu-Weather (Bi et al., 2023), leveraging Swin Transformers with earth-specific location embeddings, surpasses NWP baselines and demonstrates strong scalability. GraphCast (Lam et al., 2022) advances further by introducing message-passing networks to efficiently capture global dependencies and improve accuracy. FuXi (Chen et al., 2023b) extends the forecasting horizon, delivering 15-day predictions at skill levels comparable to ECMWF operational systems. FengWu (Chen et al., 2023a) integrates multi information flow, enabling improved representation of different atmospheric dynamics. Despite these advances, most existing models are primarily designed to learn a mapping from the current state to the next-step state, which limits their ability to capture long-range temporal evolution and synergistic cross-scale interactions.

**Spatio-Temporal Modeling** Recent advances in time-series forecasting have significantly improved modeling capacity across diverse settings, including loss-function design (Wang et al., 2026b; 2025c;b;a; 2026c; Qiu et al., 2025b), architectural innovations spanning linear, MLP-, Transformer-, and graph-based models (Wang et al., 2023b; 2026a; Yue et al., 2025; Qiu et al., 2025c; Wu et al., 2025b; Hu et al., 2025b), exogenous and irregular multivariate modeling (Li et al., 2026; Qiu et al., 2025d; Liu et al., 2026a;b), probabilistic and multimodal forecasting (Wu et al., 2025c; 2026), as well as comprehensive benchmarking efforts (Qiu et al., 2024; 2025a; Hu et al., 2025a). Building upon these developments, research has progressively extended from temporal modeling to spatio-temporal

forecasting, where spatial interactions, graph structures, and cross-scale dynamics become central modeling challenges (Ma et al., 2025c; Hu et al., 2026). Deep learning approaches for spatio-temporal modeling mainly focus on two data forms: *graph-structured* and *grid-structured* data (Liang et al., 2025). While graph-based methods have also been explored for irregular domains (Zhang et al., 2022; Wang et al., 2023a; 2024a), grid-based modeling remains particularly suitable for global-scale atmospheric data due to its structured nature. Early approaches such as ConvLSTM (Shi et al., 2015) and PredRNN (Wang et al., 2017) explicitly model temporal dependencies through recurrent connections, achieving promising results on precipitation nowcasting. To improve efficiency, SimVP (Gao et al., 2022a) replaces recurrent structures with pure convolutional operators, demonstrating competitive performance on benchmark sequence modeling tasks . More recently, Transformer-based architectures have been introduced, with models such as Earthformer (Gao et al., 2022b) leveraging attention mechanisms on gridded climate data to enhance long-range spatio-temporal dependencies. However, these models are primarily designed as general-purpose spatio-temporal learners and lack specialized adaptations to the unique multi-scale dynamics of weather and climate systems.

## 3 PROBLEM DEFINITION

In this study, we formulate weather forecasting as a multi-step spatiotemporal forecasting problem. At each time step $t$, the meteorological state consists of surface variables $\mathbf{S}_t$ and atmospheric variables $\mathbf{A}_t$ across 13 pressure levels. We concatenate them along the channel dimension to form the combined input: $\mathbf{X}_t = [\mathbf{S}_t, \mathbf{A}_t] \in \mathbb{R}^{H \times W \times C}$, where $H \times W$ is the number of spatial grid locations, and $C = C_s + C_a$ is the total number of variables, with $C_s$ and $C_a$ denoting the numbers of surface and atmospheric variables, respectively. Our formulation is based on a history of $T$ steps to directly forecast the next $L$ steps. Specifically, given historical inputs $\mathbf{X}_{t-T+1:t}$, the model predicts the future $\hat{\mathbf{X}}_{t+1:t+L} = \text{Model}(\mathbf{X}_{t-T+1:t}; \Theta)$, where $\Theta$ denotes the model parameters. To support ultra-long forecasting horizons (far beyond $L$), we adopt a recursive rollout strategy. After generating the first $L$ predictions, the most recent $T$ predicted states are fed back as inputs to produce the next block of $L$ forecasts, and this process is repeated. This block-wise multi-step forecasting significantly alleviates the error accumulation typical of purely autoregressive one-step methods, enabling stable and accurate long-horizon predictions.

## 4 METHODS

As illustrated in Fig. 2, we introduce STORM, a unified framework for multi-scale spatio-temporal forecasting. The design of STORM is motivated by the intrinsic hierarchical nature of atmospheric dynamics, where large-scale circulations and small-scale local processes interact across different temporal horizons. To capture such complex dependencies, STORM consists of three key components: (i) **Hierarchical Earth Embedder** that progressively downsamples raw observations to construct multi-resolution representations, (ii) **Scale-bridging Spatio-temporal Encoder** that jointly models temporal evolution and spatial interactions while enabling information propagation from fine- to coarse-grained levels, and (iii) **Level-Aligned Forecasting Decoder** that projects multi-scale representations into future prediction through depth-specific transposed convolutions. Together, these components allow STORM to effectively decompose, bridge, and reconstruct atmospheric dynamics across scales for accurate weather forecasting.

### 4.1 HIERARCHICAL EARTH EMBEDDER

Earth observation data inherently exhibit multi-scale characteristics: coarse scales capture large-scale circulation patterns, while fine scales reflect local variations and detailed structures. To effectively leverage information across these scales, we propose the *Hierarchical Earth Embedder* for multi-scale embedding of global meteorological features. Given input data $\mathbf{X} = [\mathbf{S}, \mathbf{A}] \in \mathbb{R}^{T \times H \times W \times C}$, where $T$ denotes the historical length, $C$ represents the total number of surface and atmospheric variables, and $H \times W$ defines a fine-grained global grid, the data contains rich local details but makes it challenging to directly capture coarse-scale circulation and large-scale dependencies. To address this, we adopt a progressive, layer-wise downsampling strategy for multi-scale modeling and embedding.

First, a $3 \times 3$ spatial convolution is applied to the original fine-grained input to obtain the initial embedding $\mathbf{H}_0 \in \mathbb{R}^{T \times H \times W \times D}$, where $D$ is the hidden embedding dimension. This step allows the

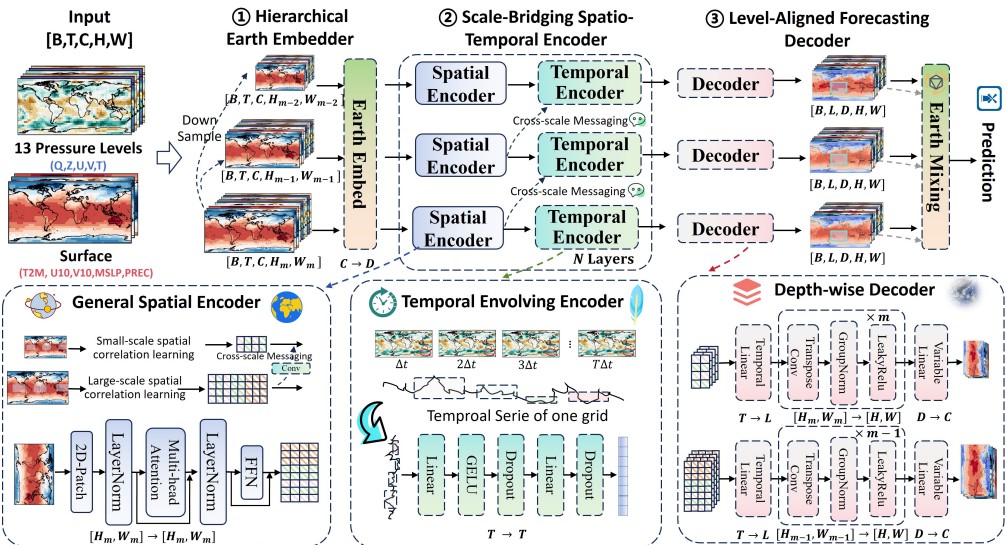

Figure 2: Illustration of the multi-scale collaborative meteorological forecasting model.

model to learn localized spatial representations while embedding the raw data. Subsequently, a series of strided convolutional layers with stride 2 are applied to progressively downsample the embeddings and capture increasingly coarser spatial structures:

$$\mathbf{H}_m = \text{LeakyReLU}(\text{GroupNorm}(\text{Conv2d}(\mathbf{H}_{m-1}; \text{kernel} = 3, \text{stride} = 2)))), \tag{1}$$

where $\text{LeakyReLU}$ denotes the activation function and $\text{GroupNorm}$ normalizes feature channels to improve training stability and convergence.

The resulting multi-scale embeddings $\mathcal{H} = \{\mathbf{H}_0, \mathbf{H}_1, \mathbf{H}_2, ..., \mathbf{H}_M\}$, where $\mathbf{H}_0 \in \mathbb{R}^{T \times \frac{H}{2^0} \times \frac{W}{2^0} \times D}$ retains fine-grained local representations and $\mathbf{H}_M \in \mathbb{R}^{T \times \frac{H}{2^M} \times \frac{W}{2^M} \times D}$ encodes coarse-scale global structures, capture multi-scale spatio-temporal features. These embeddings are then fed into the *Scale-bridging Spatio-Temporal Encoder*.

## 4.2 SCALE-BRIDGING SPATIO-TEMPORAL ENCODER

After obtaining the multi-scale Earth representations, we design a scale-bridging spatio-temporal encoder to jointly capture spatial dependencies, temporal dynamics, and inter-scale interactions. Unlike traditional meteorological models that often emphasize either temporal or spatial aspects in isolation, our encoder integrates both perspectives while enabling information flow across scales, which is critical for representing hierarchical atmospheric processes.

**Spatial encoding.** Given multi-scale features $\mathcal{H}$, each level is first processed by an independent spatial encoder inspired by the Vision Transformer (Dosovitskiy et al.) structure. Specifically, for the $m$-th scale and the $n$-th encoder layer, we perform 2D patch embedding, followed by multi-head self-attention and feed-forward transformations:

$$\mathbf{H}_m^{s'(n)} = \text{LayerNorm}(\text{Patch2D}(\mathbf{H}_m^{(n)})) + \text{MHA}(\text{LayerNorm}(\text{Patch2D}(\mathbf{H}_m^{(n)}))), \tag{2}$$

$$\mathbf{H}_m^{s(n)} = \text{DePatch2D}\left(\text{LayerNorm}(\mathbf{H}_m^{s'(n)}) + \text{FFN}(\text{LayerNorm}(\mathbf{H}_m^{s'(n)}))\right), \tag{3}$$

where $\text{Patch2D}$ partitions the grid of size $\frac{H}{2^m} \times \frac{W}{2^m}$ into $(P_1, P_2)$ patches, producing $\frac{H \times W}{P_1 \times P_2}$ tokens, and $\text{DePatch2D}$ restores the tokenized representation back to its grid structure. This operation yields the spatial encoding $\mathbf{H}_m^{s(n)} \in \mathbb{R}^{T \times \frac{H}{2^m} \times \frac{W}{2^m} \times D}$, which captures scale-specific meteorological spatial dependencies.

**Cross-scale message passing.** To enhance coherence between adjacent scales, we introduce a cross-scale messaging mechanism. Information from the finer resolution is downsampled and injected into the coarser scale, facilitating hierarchical knowledge transfer:

$$\mathbf{H}_m^{c(n)} = \text{CrossScaleMessaging}(\mathbf{H}_{m-1}^{s(n)}) + \mathbf{H}_m^{s(n)}. \tag{4}$$

Here, CrossScaleMessaging is implemented as a stride-2 convolution, which aligns the resolution of $\mathbf{H}_{m-1}^{s(n)}$ with $\mathbf{H}_m^{s(n)}$. This design enables the encoder to integrate localized fine-grained patterns into broader global representations, crucial for multi-scale weather forecasting.

**Temporal encoding.** Finally, we model the temporal evolution of the cross-scale features $\mathbf{H}_m^{c(n)}$. While self-attention could be applied across temporal tokens, the extremely high dimensionality of meteorological grids makes it computationally prohibitive. Motivated by recent findings in long-term time series forecasting, we adopt a lightweight linear temporal encoder that achieves stable temporal modeling with minimal overhead:

$$\mathbf{H}_m^{t(n)} = W_2 \times \text{GELU}(W_1 \times \mathbf{H}_m^{s(n)} + B_1) + B_2 \tag{5}$$

This lightweight network is used to extract temporal dependencies. Here, $W_1 \in \mathbb{R}^{T \times 2T}$ and $W_2 \in \mathbb{R}^{T \times 2T}$, enabling effective modeling of historical time steps across multiple scales with an extremely small number of parameters (fewer than 100). It efficiently captures multi-scale temporal relationships in past observations.

### 4.3 Level-Aligned Forecasting Decoder

The outputs from the $N$-layer scale-bridging spatio-temporal encoder are subsequently fed into a level-aligned forecasting decoder to generate multi-scale forecasts of future atmospheric states. The decoder is designed to progressively reconstruct high-resolution spatio-temporal fields from the encoded representations, while ensuring that temporal forecasting and spatial upsampling remain coherent across scales. For each resolution level $m$, we apply an independent temporal linear layer that maps the $T$-step historical features into $L$-step future predictions. This design explicitly decouples temporal forecasting across scales, enabling each level to capture temporal dynamics consistent with its own resolution:

$$\mathbf{Z}_m = \text{Linear}_m(\mathbf{H}_m^{t(N)}) \in \mathbb{R}^{L \times \frac{H}{2^m} \times \frac{W}{2^m} \times D}. \tag{6}$$

To progressively upsample the spatial resolution, we adopt a depth-adaptive transposed convolution block. Each block doubles the resolution along both spatial dimensions, and the depth of upsampling is matched to the level index $m$. Thus, the $m$-th scale undergoes $m$ successive upsampling layers, ensuring that coarse-scale features are refined consistently into finer spatial grids:

$$\mathbf{U}_m = \text{DeConv}^m(\mathbf{Z}_m) \in \mathbb{R}^{L \times H \times W \times D}. \tag{7}$$

This hierarchical upsampling ensures that global context encoded at coarse scales is gradually transformed into high-resolution local forecasts.

After spatial refinement, all scales share a common variable-wise linear projection layer, which maps the hidden dimension $D$ into the full set of meteorological variables, including both surface and atmospheric fields. This shared mapping enforces consistency across scales and guarantees that all outputs lie in the same physical variable space:

$$\mathbf{V}_m = \text{Linear}_{var}(\mathbf{U}_m) \in \mathbb{R}^{L \times H \times W \times C}. \tag{8}$$

Finally, the forecasts from all scales are integrated to form the unified prediction. Each level provides complementary information, with fine-scale forecasts capturing localized details and coarse-scale forecasts contributing global stability. In EarthMix, this integration is performed by directly summing the scale-specific representations:

$$\hat{\mathbf{X}} = \text{EarthMix}\left(\{\mathbf{V}_m\}_{m=1}^M\right) \in \mathbb{R}^{L \times H \times W \times C}. \tag{9}$$

This simple aggregation efficiently combines multi-scale information to produce the final high-resolution, multi-variable prediction.

## 5 EXPERIMENTS

**Dataset.** We conduct experiments on the fifth generation of ECMWF Reanalysis data (ERA5) (Rasp et al., 2023). The temporal range spans from 1993 to 2021, where 1993–2017 is used for training, 2018–2019 for validation, and 2020–2021 for testing. We consider both atmospheric and surface variables. The atmospheric variables include five pressure-level quantities, each with 13 pressure levels: geopotential (Z), specific humidity (Q), temperature (T), and the U and V components of wind speed. The surface variables include 10-meter wind components (U10M, V10M), 2-meter temperature (T2M), total precipitation, and mean sea-level pressure (MSLP). To comprehensively evaluate our method, we construct three datasets at different spatial scales: ❶ **Global-level:** We use the preprocessed ERA5 dataset with a $5.625°$ spatial resolution and 6-hour temporal frequency. ❷ **Continental-level:** We extract ERA5 data over South America, covering the region from $56°$S to $14°$N and $81°$W to $34°$W, with a $1°$ spatial resolution. ❸ **Regional-level:** We extract a subset with $0.25°$ resolution over East Asia ($20°$N–$28°$N, $110°$E–$126°$E). More data and pre-processsing details are available in Appendix C

**Metrics.** To evaluate forecasting performance, we report two widely adopted metrics in numerical weather prediction: latitude-weighted Root Mean Squared Error (RMSE) and Anomaly Correlation Coefficient (ACC). All predictions are first de-normalized before metric computation. Given $L$ forecasting steps, the prediction $\hat{x}_{\ell hw}$ and ground truth $y_{\ell hw}$ at lead time $\ell$ and spatial grid $(h, w)$, the metrics are defined as

$$\text{RMSE} = \frac{1}{L} \sum_{\ell=1}^{L} \sqrt{\frac{1}{HW} \sum_{h=1}^{H} \sum_{w=1}^{W} \alpha(h) \left(y_{\ell hw} - \hat{x}_{\ell hw}\right)^2}, \text{ACC} = \frac{\sum_{\ell,h,w} \alpha(h)\, \tilde{y}_{\ell hw}\, \tilde{\hat{x}}_{\ell hw}}{\sqrt{\sum_{\ell,h,w} \alpha(h)\, \tilde{y}_{\ell hw}^2} \sqrt{\sum_{\ell,h,w} \alpha(h)\, \tilde{\hat{x}}_{\ell hw}^2}},$$
(10)

where the latitude-dependent weight is $\alpha(h) = \cos(h)/\frac{1}{H} \sum_{h'=1}^{H} \cos(h')$, and $\tilde{y} = y - C$, $\tilde{\hat{x}} = \hat{x} - C$ are anomalies relative to the empirical climatology $C = \frac{1}{LHW} \sum_{\ell,h,w} y_{\ell hw}$. More Details is in Appendix D.

**Baselines.** We benchmark our approach against a diverse set of representative baselines spanning operational weather forecasting models, spatio-temporal sequence learners, and image-based architectures. For single-step meteorological forecasting, we include **Pangu-Weather** (Bi et al., 2023), **FuXi** (Chen et al., 2023b), **FourCastNet (FCN)** (Pathak et al., 2022), and **Triton** (Wu et al., 2025a), which represent state-of-the-art Transformer-based designs for data-driven numerical weather prediction. For multi-step spatio-temporal modeling, we evaluate against **SimVP** (Gao et al., 2022a), a leading video prediction framework adapted for geophysical data. Additionally, we incorporate a canonical vision backbone, **U-Net** (Ronneberger et al., 2015), to assess the performance of convolutional architectures in this setting. All baselines are re-trained on our datasets with identical variables and splits to ensure fairness of comparison. More Details can be found in Appendix E.

### 5.1 GLOBAL WEATHER FORECASTING

Table 1: Quantitative comparison of short-term (up to 24 hours) global weather forecasting performance. Metrics are reported as weighted RMSE and ACC, averaged over $\Delta t = \{6, 12, 18, 24\}$ hours.

| Metric | | | | RMSE↓ | | | | | | | ACC↑ | | | |
|---|---|---|---|---|---|---|---|---|---|---|---|---|---|---|
| Variable | STORM | Triton | Pangu | FCN | Fuxi | SimVP | UNet | STORM | Triton | Pangu | FCN | Fuxi | SimVP | UNet |
| T2M | **0.675** | 0.873 | 1.106 | 1.579 | 1.356 | 1.343 | 2.445 | **0.999** | 0.998 | 0.997 | 0.994 | 0.995 | 0.996 | 0.986 |
| U10 | **0.669** | 0.819 | 1.377 | 1.529 | 1.314 | 1.448 | 1.971 | **0.991** | 0.987 | 0.959 | 0.953 | 0.965 | 0.959 | 0.952 |
| V10 | **0.713** | 0.868 | 1.483 | 1.658 | 1.381 | 1.556 | 1.867 | **0.984** | 0.976 | 0.924 | 0.910 | 0.940 | 0.923 | 0.915 |
| Prec | **5.4E-04** | 6.4E-04 | 8.4E-04 | 9.8E-04 | 8.7E-04 | 9.6E-04 | 1.0E-03 | **0.923** | 0.891 | 0.802 | 0.728 | 0.791 | 0.744 | 0.765 |
| MSLP | **71.8** | 93.7 | 215.8 | 217.1 | 170.1 | 185.8 | 257.5 | **0.998** | 0.996 | 0.978 | 0.979 | 0.988 | 0.986 | 0.976 |
| U500 | **1.903** | 2.500 | 3.410 | 4.607 | 3.962 | 4.066 | 4.677 | **0.993** | 0.988 | 0.975 | 0.958 | 0.969 | 0.968 | 0.963 |
| V500 | **2.055** | 2.597 | 3.709 | 5.021 | 4.160 | 4.380 | 4.999 | **0.983** | 0.974 | 0.942 | 0.899 | 0.934 | 0.926 | 0.914 |
| T500 | **0.524** | 0.696 | 0.886 | 1.251 | 1.111 | 1.242 | 1.237 | **0.998** | 0.997 | 0.995 | 0.991 | 0.993 | 0.991 | 0.991 |
| Z500 | **79.8** | 109.4 | 225.7 | 268.5 | 215.3 | 223.5 | 308.3 | **1.000** | 1.000 | 0.998 | 0.998 | 0.999 | 0.999 | 0.997 |
| Q500 | **3.9E-05** | 5.0E-05 | 5.6E-05 | 7.3E-05 | 6.9E-05 | 7.7E-05 | 7.3E-05 | **0.976** | 0.962 | 0.949 | 0.918 | 0.926 | 0.909 | 0.924 |

**Short-term Forecasting.** We first evaluate the proposed multi-scale spatiotemporal forecasting model, STORM, on short-term global weather prediction. Specifically, we conduct 24-hour forecasts and compare our method against a range of state-of-the-art approaches, including Pangu, Fuxi, Four-CastNet, Triton, SimVP, and U-Net. The evaluation covers both surface variables (2m temperature (T2M), 10m winds (U10, V10), precipitation (PREC), and mean sea level pressure (MSLP)) as well as mid-tropospheric fields (500 hPa U/V winds, temperature (T), geopotential height (Z), and specific humidity (Q)). For each method, we report weighted RMSE and ACC averaged over the prediction horizons $\Delta t = \{6, 12, 18, 24\}$ hours. The results, summarized in Table 1, show that STORM consistently achieves the best performance across all variables. In terms of RMSE, our method surpasses even Triton, a model particularly tailored for short-term forecasting, by a significant margin, while also delivering superior ACC scores. These improvements highlight the advantage of incorporating multi-scale spatiotemporal representations, which enable the model to capture both local details and global circulation patterns effectively. Full results are available at Appendix J.

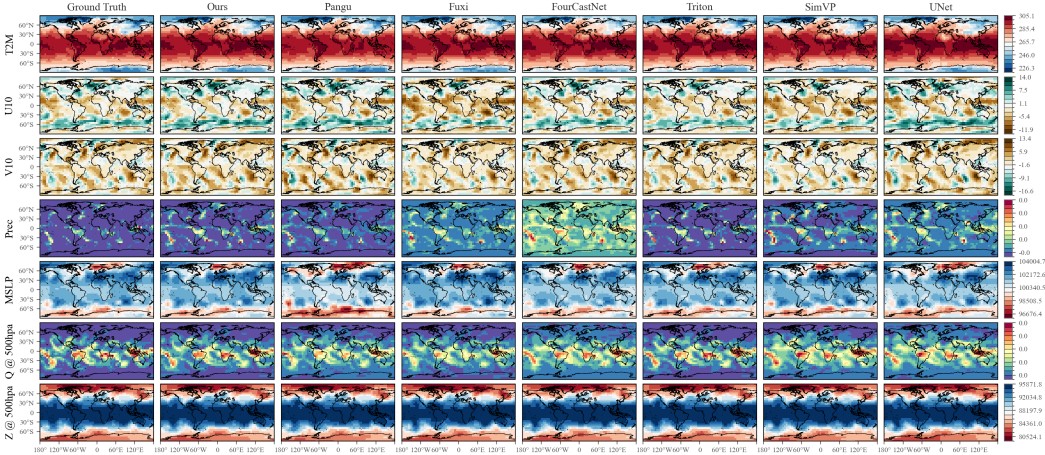

Figure 3: Visualization of 24-hour forecasts for multiple atmospheric variables.

To further illustrate the predictive capability of our model, Figure 3 visualizes 24-hour forecasts across multiple variables. Compared with other baselines, the predictions of STORM are visually closer to the ground-truth reanalysis, preserving fine-grained regional structures while maintaining coherent large-scale dynamics. These results confirm that our approach not only improves numerical accuracy but also produces more realistic spatial patterns in short-term global weather forecasting. The visualizations for all variables are provided in Appendix I.

Table 2: Quantitative comparison of long-term (7–10 days) global weather forecasting performance. Metrics are reported as weighted RMSE and ACC, averaged over $\Delta t = \{168, 192, 216, 240\}$ hours.

| Metric | | | RMSE↓ | | | | | | | ACC↑ | | | | |
|---|---|---|---|---|---|---|---|---|---|---|---|---|---|---|
| Variable | STORM | Triton | Pangu | FCN | Fuxi | SimVP | UNet | STORM | Triton | Pangu | FCN | Fuxi | SimVP | UNet |
| T2M | **2.596** | 3.386 | 3.633 | 4.181 | 3.500 | 3.168 | 4.477 | **0.984** | 0.965 | 0.955 | 0.971 | 0.969 | 0.976 | 0.937 |
| U10 | **3.857** | 4.271 | 5.127 | 5.363 | 4.696 | 3.901 | 5.828 | **0.716** | 0.650 | 0.615 | 0.604 | 0.574 | 0.684 | 0.559 |
| V10 | **4.106** | 4.504 | 5.095 | 5.569 | 4.795 | 4.180 | 6.045 | **0.464** | 0.370 | 0.325 | 0.297 | 0.282 | 0.383 | 0.262 |
| Prec | **1.4E-03** | 2.1E-03 | 1.8E-03 | 1.6E-03 | 1.6E-03 | 1.5E-03 | 2.3E-03 | **0.432** | 0.305 | 0.288 | 0.223 | 0.243 | 0.382 | 0.204 |
| MSLP | **735.4** | 815.6 | 1125.4 | 1041.4 | 950.0 | 784.7 | 1149.3 | **0.782** | 0.725 | 0.741 | 0.718 | 0.641 | 0.741 | 0.679 |
| U925 | **6.582** | 8.918 | 10.794 | 9.159 | 10.152 | 8.117 | 12.236 | **0.884** | 0.787 | 0.803 | 0.793 | 0.747 | 0.827 | 0.747 |
| V925 | **6.588** | 7.910 | 9.707 | 9.618 | 8.794 | 7.148 | 10.858 | **0.519** | 0.369 | 0.299 | 0.284 | 0.212 | 0.336 | 0.205 |
| T925 | **2.797** | 3.718 | 4.308 | 3.879 | 4.513 | 3.565 | 5.331 | **0.966** | 0.937 | 0.931 | 0.938 | 0.912 | 0.945 | 0.902 |
| Q925 | **3.3E-07** | 5.4E-07 | 4.5E-07 | 4.7E-07 | 5.5E-07 | 4.0E-07 | 6.3E-07 | **0.749** | 0.556 | 0.556 | 0.572 | 0.535 | 0.630 | 0.519 |
| Z925 | **813.4** | 1049.4 | 1419.2 | 968.8 | 1304.1 | 1097.4 | 1537.9 | **0.981** | 0.959 | 0.955 | 0.963 | 0.958 | 0.971 | 0.936 |

**Long-term Forecasting.** Beyond short horizons, we further evaluate STORM on extended-range forecasting, covering 7-day, 8-day, 9-day, and 10-day predictions. Table 2 reports the average weighted RMSE and ACC across these horizons. The results show that STORM consistently outperforms all competing methods, with clear margins over strong baselines such as Pangu and

FourCastNet. Even at day 10, where the forecasting task becomes extremely challenging due to error accumulation and chaotic dynamics, our model maintains significantly lower RMSE and higher ACC, demonstrating its robustness and superior long-term predictive capability. Full results are available at Appendix K. To provide further insight, Figure 4 visualizes the evolution of RMSE and ACC from

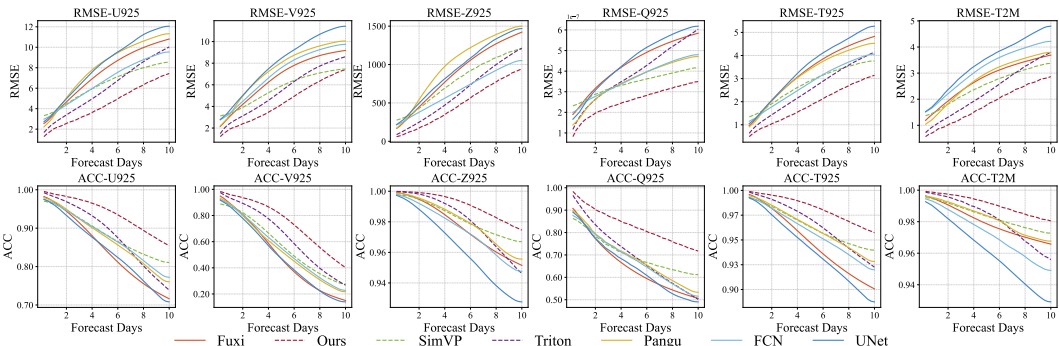

Figure 4: RMSE and ACC of U925, V925, Z925, Q925, T925, and T2M from day 1 to day 10.

day 1 to day 10 for several key variables, including U925, V925, Z925, Q925, T925, and T2M. We observe that different baseline methods tend to exhibit strengths over specific forecast horizons—for example, some achieve relatively competitive results at shorter lead times but deteriorate rapidly afterwards. In contrast, STORM consistently maintains the best accuracy across both short- and long-term horizons. This highlights the effectiveness of our multi-scale spatiotemporal design, which enables stable and reliable forecasts over a wide range of temporal scales.

## 5.2 REGIONAL HIGH RESOLUTION FORECASTING

To further demonstrate the scalability and robustness of our approach, we conduct forecasting experiments on higher-resolution regional subsets. Specifically, we consider two representative cases: Continental-level: South America, spanning from 56°S to 14°N and 81°W to 34°W, at 1° resolution; Regional-level: East Asia, spanning 20°N–28°N and 110°E–126°E, at 0.25° resolution.

Table 3 summarizes the RMSE and ACC across all variables for horizons of 6 hours, 1 day, 4 days, 7 days, and 10 days. We observe that STORM consistently achieves the lowest RMSE and the highest ACC at both continental and regional scales, highlighting its ability to maintain strong predictive skill under different spatial resolutions. In particular, the advantage of STORM becomes more pronounced at longer horizons, demonstrating its superiority in long-range regional forecasting. Figure 5 provides a qualitative visualization of the 10-day forecasts for selected variables in both South America and East Asia. By zooming into local details, we see that STORM captures fine-grained structures more faithfully and is visually closer to the ground truth compared to baselines.

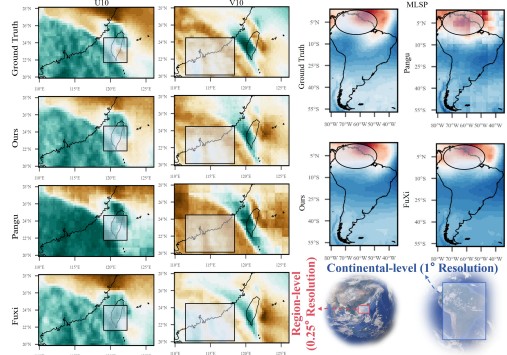

Figure 5: Visualization of 10-day forecasts over South America (1°) and East Asia (0.25°).

These results confirm the effectiveness of our multi-scale design in enhancing both large-scale consistency and small-scale detail preservation.

## 5.3 ABLATION STUDY

To better understand the contribution of each component in STORM, we conduct an ablation study by removing or modifying key modules. Specifically, we consider the following variants, ❶ **w/o T**: which removes the temporal evolution encoder from the encoder; ❷ **w/o M**: which changes the Hierarchical Earth Embedder, the Scale-Bridging Spatio-Temporal Encoder, and the Level-Aligned Forecasting Decoder to replace multi-scale modeling with single-scale modeling; ❸ **w/o S**: which removes the spatial encoder from the encoder.

Table 3: Regional high-resolution forecasting results. We report averaged results for 6 hours, 1 day, 4 days, 7 days, and 10 days over South America (1° resolution) and East Asia (0.25° resolution).

| Data | Methods | 6Hour | | 1Day | | 4Day | | 7Day | | 10Day | |
|---|---|---|---|---|---|---|---|---|---|---|---|
| | | RMSE↓ | ACC↑ | RMSE↓ | ACC↑ | RMSE↓ | ACC↑ | RMSE↓ | ACC↑ | RMSE↓ | ACC↑ |
| Cont-SA (1°) | STORM | **0.874** | **0.945** | **1.560** | **0.847** | **2.866** | **0.601** | **3.322** | **0.519** | **3.510** | **0.499** |
| | Triton | 0.927 | 0.941 | 1.808 | 0.817 | 3.689 | 0.480 | 4.687 | 0.347 | 5.433 | 0.251 |
| | FCN | 1.411 | 0.933 | 2.466 | 0.816 | 4.186 | 0.498 | 5.176 | 0.425 | 6.018 | 0.403 |
| | Pangu | 0.989 | 0.876 | 1.794 | 0.702 | 3.557 | 0.440 | 4.235 | 0.375 | 4.613 | 0.368 |
| | Fuxi | 0.913 | 0.941 | 1.677 | 0.830 | 3.253 | 0.547 | 3.632 | 0.500 | 3.797 | 0.466 |
| | SimVP | 0.947 | 0.939 | 1.619 | 0.843 | 3.110 | 0.554 | 3.603 | 0.476 | 3.891 | 0.426 |
| | UNet | 1.528 | 0.873 | 2.696 | 0.661 | 6.994 | 0.291 | 11.50 | 0.248 | 15.62 | 0.252 |
| Reg-EA (0.25°) | STORM | **14.53** | **0.980** | **41.71** | **0.920** | **154.6** | **0.627** | **187.8** | **0.548** | **200.9** | **0.524** |
| | Triton | 14.55 | 0.976 | 51.65 | 0.919 | 183.7 | 0.579 | 218.7 | 0.499 | 234.3 | 0.482 |
| | FCN | 33.00 | 0.954 | 98.15 | 0.832 | 256.4 | 0.559 | 297.4 | 0.495 | 315.6 | 0.471 |
| | Pangu | 17.35 | 0.963 | 47.31 | 0.895 | 157.5 | 0.625 | 206.2 | 0.543 | 227.5 | 0.511 |
| | Fuxi | 17.69 | 0.969 | 54.67 | 0.903 | 178.1 | 0.592 | 211.2 | 0.522 | 227.7 | 0.493 |
| | SimVP | 32.17 | 0.935 | 53.44 | 0.887 | 158.5 | 0.615 | 189.0 | 0.546 | 202.5 | 0.523 |
| | UNet | 22.77 | 0.956 | 63.05 | 0.875 | 195.1 | 0.566 | 241.6 | 0.480 | 261.8 | 0.450 |

Figure 6 reports the RMSE for ten key variables: T2M, U10, V10, PREC, MSLP, U500, V500, T500, Q500, and Z500. The results demonstrate several important findings: (1) Every module contributes significantly to the overall forecasting performance; (2) The spatial encoder (**S**) plays a particularly critical role in improving pre-

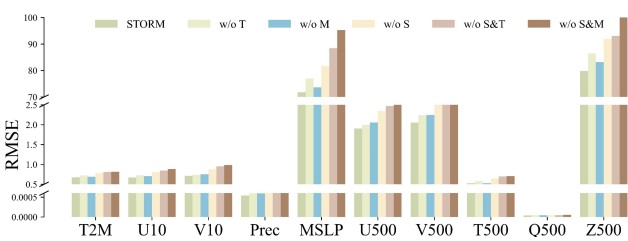

Figure 6: Ablation study results of STORM.

diction accuracy; (3) The combination of multi-scale mixing and spatial encoding achieves the largest performance gains, indicating that capturing both spatial dependencies and multi-scale interactions is crucial for accurate global weather forecasting. These observations confirm the effectiveness of our design in modeling complex spatiotemporal dependencies inherent in meteorological data.

## 5.4 STORM ANALYSIS

We perform a multi-scale analysis of STORM to understand the contribution of different spatial scales to forecasting accuracy. Figure 7(a) visualizes predictions at different scales. The finest scale (scale0) captures rich local details, while the coarsest scale (scale2) better represents global circulation patterns. Individually, each scale exhibits distinct prediction errors, but combining multi-scale predictions results in the most accurate forecasts, demonstrating the effectiveness of multi-scale integration. Figure 7(b) further analyzes the effect of the number of scales as a hyperparameter. As the number of scales increases, the predictive performance improves, but the gains diminish beyond three scales. Based on this analysis, we set the number of scales in STORM to three, achieving a balance between local detail preservation and global structure representation.

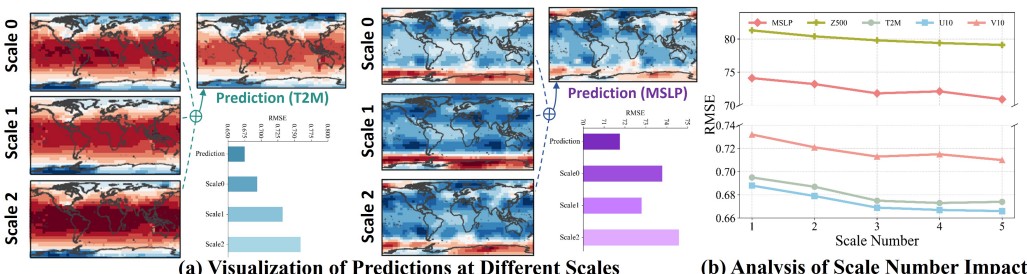

(a) Visualization of Predictions at Different Scales  (b) Analysis of Scale Number Impact

Figure 7: Visualization of multi-scale predictions and the analysis of scale number.

## 6 CONCLUSION

Despite rapid advances in deep learning for weather prediction, modeling atmospheric dynamics remains challenging due to their heterogeneous spatio-temporal scales. We introduced **STORM**, a unified framework with a Hierarchical Earth Embedder, a Scale-Bridging Spatio-Temporal Encoder, and a Level-Aligned Forecasting Decoder to explicitly capture cross-scale dependencies. Extensive experiments on ERA datasets with resolutions of $5.625°$, $1°$, and $0.25°$ show that STORM consistently achieves state-of-the-art performance across global and regional settings, as well as short- and long-term horizons. Our results highlight the necessity of synergistic multi-scale modeling for reliable weather forecasting. Looking ahead, we envision STORM as a step toward a new generation of spatio-temporal models that integrate fine-grained detail with global coherence, offering a foundation for more accurate, efficient, and robust climate and Earth system prediction.

## ETHICS STATEMENT

All authors confirm compliance with the ICLR Code of Ethics. This study does not utilize human participants, private information, or sensitive content. Our work relies entirely on publicly available datasets and established benchmarks, with no anticipated negative social or environmental consequences. There are no conflicts of interest or external funding sources that might influence the findings.

## REPRODUCIBILITY STATEMENT

We have taken care to ensure that our experiments can be replicated. The architecture and training details of our model are fully described in Sections 4 and Appendix E. All datasets are publicly accessible, and the preprocessing steps are documented in Appendix C. Furthermore, we provide anonymized code and usage instructions as supplementary material to support reproducibility of our results.

## ACKNOWLEDGMENT

This paper is partially supported by the National Natural Science Foundation of China (No.62502488, No.12227901), Natural Science Foundation of Jiangsu Province (BK20240460), the grant from State Key Laboratory of Resources and Environmental Information System. The AI-driven experiments, simulations and model training were performed on the robotic AI-Scientist platform of Chinese Academy of Sciences.

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

# Appendix

## Table of Contents

## A LLM USAGE

Following the conference guidelines regarding large language models (LLMs), we disclose that LLMs were utilized solely to improve sentence clarity and grammatical correctness. All aspects of conceptual development, experimental methodology, data analysis, and core manuscript content were independently produced by the authors without LLM assistance

## B THEORETICAL ANALYSIS: WHY MULTI-SCALE MODELING HELPS

In this section we present theoretical results that explain why a synergistic cross-scale spatio-temporal architecture can outperform a single-scale counterpart in (i) statistical generalization (smaller expected prediction error) and (ii) optimization speed (faster convergence). We adopt a bias–variance decomposition perspective for generalization, and a condition-number / PL-type argument for optimization. Proofs are sketched; full technical details follow the argument outlines below.

### B.1 SETUP AND ASSUMPTIONS

Let $\mathbf{X} = \mathbb{R}^{T \times C \times H \times W}$ denote one time-slice grid and consider a target spatio-temporal mapping $f^\star : \mathbb{R}^{T \times C \times H \times W} \to \mathbb{R}^{L \times C \times H \times W}$ that maps $T$ historical frames to $L$ future frames (all variables vectorized as needed). Denote by $\mathcal{D}$ the data distribution and by $(\mathbf{X}_i, Y_i)_{i=1}^n \sim \mathcal{D}$ the training set. We consider two model classes:**❶ Single-scale model** $\mathcal{F}_{\text{single}}$ of effective parameter-dimension $d$ (e.g., a monolithic spatio-temporal network operating at full resolution). **❷ Multi-scale model** $\mathcal{F}_{\text{ms}}$ which decomposes the prediction into $M$ scale-specific modules with parameter-dimensions $d_1, \ldots, d_M$ (so $\sum_{m=1}^M d_m = d$ or $\leq d$ depending on parameter sharing). We make standard regularity assumptions:

**Assumption B.1** (Decomposability). *The target admits a scale decomposition*

$$f^\star = \sum_{m=1}^M f_m^\star, \qquad f_m^\star \in \mathcal{G}_m,$$

*where each $f_m^\star$ captures variations at spatial scale $m$, and $\mathcal{G}_m$ is the function space for scale $m$ signals (e.g., fine, medium, or coarse patterns).*

**Assumption B.2** (Model capacity allocation). *Each scale-model class $\mathcal{F}_m$ has Rademacher complexity $\mathfrak{R}_n(\mathcal{F}_m) \leq \rho_m \sqrt{\frac{d_m}{n}}$ for constants $\rho_m > 0$, while the single-scale class satisfies $\mathfrak{R}_n(\mathcal{F}_{\text{single}}) \leq \rho \sqrt{\frac{d}{n}}$.*

**Assumption B.3** (PL condition for optimization). *The empirical loss $\hat{L}(\theta)$ is $L$-smooth and satisfies the Polyak–Łojasiewicz (PL) inequality in neighborhoods of interest: there exists $\mu > 0$ s.t.*

$$\frac{1}{2} \|\nabla \hat{L}(\theta)\|^2 \geq \mu (\hat{L}(\theta) - \hat{L}^\star).$$

### B.2 GENERALIZATION BOUND IMPROVEMENT

We first bound the excess risk (population loss minus Bayes risk) via a bias–variance decomposition and Rademacher complexity.

**Theorem B.1** (Generalization bound imporvement). *Assume squared loss and hypotheses above. Let $\hat{f}_{\text{single}}$ and $\hat{f}_{\text{ms}} = \sum_{m=1}^M \hat{f}_m$ be ERM solutions in $\mathcal{F}_{\text{single}}$ and $\mathcal{F}_{\text{ms}}$ respectively. Then with probability at least $1 - \delta$,*

$$\mathcal{E}(\hat{f}_{\text{single}}) \leq \underbrace{\inf_{f \in \mathcal{F}_{\text{single}}} \|f - f^\star\|^2}_{\text{approx.}} + B \rho \sqrt{\frac{d}{n}} + O\left(\sqrt{\frac{\log(1/\delta)}{n}}\right),$$

*and*

$$\mathcal{E}(\hat{f}_{\text{ms}}) \leq \underbrace{\sum_{m=1}^M \inf_{g \in \mathcal{F}_m} \|g - f_m^\star\|^2}_{\text{multi-scale approx.}} + B \sum_{m=1}^M \rho_m \sqrt{\frac{d_m}{n}} + O\left(\sqrt{\frac{\log(1/\delta)}{n}}\right),$$

*where $B > 0$ is a universal constant and $\mathcal{E}$ denotes population mean squared error.*

*Proof sketch.* Standard decomposition: population risk = approximation error + estimation error. Estimation error is controlled by Rademacher complexity; using Assumption 2 we get the stated $\rho\sqrt{d/n}$ (single) and sum of $\rho_m\sqrt{d_m/n}$ (multi-scale). The remainder term follows from concentration (Talagrand / McDiarmid), yielding the $\sqrt{\log(1/\delta)/n}$ term. $\square$

**Interpretation.** If (i) the decomposition is faithful so that each $f_m^\star$ is well-approximated by $\mathcal{F}_m$ (small multi-scale approximation error), and (ii) the per-scale capacities $d_m$ concentrate (e.g. $d_m \ll d$), then

$$\sum_{m=1}^{M} \sqrt{d_m} \ll \sqrt{d},$$

hence the multi-scale estimation term $\sum_m \rho_m\sqrt{d_m/n}$ can be substantially smaller than $\rho\sqrt{d/n}$, yielding better generalization. Concretely, if $d_m \approx d/M$ and $\rho_m \approx \rho$, then

$$\sum_{m=1}^{M} \sqrt{d_m} = M\sqrt{\frac{d}{M}} = \sqrt{Md} < \sqrt{d} \quad \text{iff } M < 1,$$

so naive equal partition does not help; however, real atmospheric signals are *compressible*: coarse scales require tiny $d_m$ and only few fine-scale components need larger $d_m$, making $\sum_m \sqrt{d_m} \ll \sqrt{d}$ in practice. Thus the bound formalizes a *bias-variance tradeoff* where multi-scale modeling reduces variance without materially increasing approximation bias.

## B.3 Improved optimization speed regarding convergency

We now sketch how hierarchical (coarse-to-fine) architectures improve optimization by reducing effective condition numbers and enabling faster gradient-based convergence.

**Theorem B.2** (Improved optimization speed regarding convergency)**.** *Under Assumption 3 (PL inequality and $L$-smoothness) consider gradient descent with step size $\eta \leq 1/L$. Let $\kappa_{\text{single}} = L/\mu$ denote the condition number for the single-scale parameterization, and let $\kappa_{\text{ms}}$ be the effective condition number for a multi-scale architecture that first fits coarse parameters and then refines fine parameters (blockwise parameterization). If the cross-scale coupling operator has spectral gap $\gamma \in (0, 1)$, then*

$$\kappa_{\text{ms}} \leq (1 - \gamma)\,\kappa_{\text{single}},$$

*and gradient descent on $\mathcal{F}_{\text{ms}}$ converges linearly as*

$$\hat{L}(\theta_t) - \hat{L}^\star \leq \left(1 - \eta\mu(1 - \gamma)\right)^t \left(\hat{L}(\theta_0) - \hat{L}^\star\right).$$

*Proof sketch.* Block-partition the parameter vector $\theta = [\theta_{\text{coarse}}, \theta_{\text{fine}}]$. The Hessian $H$ of the empirical loss can be written in block form; coarse-to-fine structure makes the off-diagonal blocks small relative to diagonal blocks due to localization (this is the spectral gap $\gamma$). Using matrix perturbation bounds (Weyl-type inequalities) one shows the largest-to-smallest eigenvalue ratio of $H$ is reduced by factor $(1 - \gamma)$. Under PL, GD attains linear rate with factor $1 - \eta\mu_{\text{eff}}$, where $\mu_{\text{eff}} = \mu(1 - \gamma)$. $\square$

**Remarks.** The spectral-gap condition formalizes the intuition that coarse-scale variables capture low-frequency, high-energy components and are weakly coupled to many fine-scale modes; explicit coarse-to-fine parametrization reduces ill-conditioning caused by high-frequency components and thus accelerates optimization.

## B.4 Putting pieces together: overall error and sample complexity

Combining generalization and optimization insights yields a unified statement.

**Corollary B.3** (Sample-complexity advantage). *Suppose the decomposition is faithful and the optimizer attains an $\epsilon$-accurate empirical minimizer in $T(\epsilon)$ iterations (GD linear convergence as above). Then to achieve population error $\mathcal{E} \leq \varepsilon$, the multi-scale model requires*

$$n_{\mathrm{ms}} = \widetilde{O}\Big( \frac{1}{\varepsilon^2} \Big( \sum_{m=1}^{M} \rho_m \sqrt{d_m} \Big)^2 \Big)$$

*samples, while the single-scale model requires*

$$n_{\mathrm{single}} = \widetilde{O}\Big( \frac{1}{\varepsilon^2} \rho^2 d \Big).$$

*If $\sum_m \sqrt{d_m} \ll \sqrt{d}$, then $n_{\mathrm{ms}} \ll n_{\mathrm{single}}$.*

## C    MORE DATA DETAILS

To provide a more comprehensive description of the experimental setup, we summarize the datasets used in this work in Table 4. As introduced in the main text, all datasets are derived from the ERA5 reanalysis, covering the period from 1993 to 2021. Specifically, we split the data into 1993–2017 for training, 2018–2019 for validation, and 2020–2021 for testing. For atmospheric variables, we include five pressure-level quantities, geopotential (Z), specific humidity (Q), temperature (T), and the U and V components of wind, each defined on 13 standard pressure levels. For surface variables, we consider 10-meter wind components (U10M, V10M), 2-meter temperature (T2M), mean sea-level pressure (MSLP), and accumulated precipitation. Note that in Global- and Continental-level settings, precipitation is additionally included compared to the original 69 ERA5 variables, while in the Regional-level dataset only near-surface variables (T2M, U10M, V10M) are used. Regarding spatial coverage, we construct three datasets with different geographical ranges and resolutions: (i) **Global-level:** $5.625°$ resolution with 6-hour temporal frequency; (ii) **Continental-level:** $1°$ resolution over South America ($56°$S–$14°$N, $81°$W–$34°$W); and (iii) **Regional-level:** $0.25°$ resolution over East Asia ($20°$N–$28°$N, $110°$E–$126°$E). For preprocessing, we standardize each variable using statistics (mean and standard deviation) computed from the training set only. During inference, model predictions are rescaled back (de-normalized) to the original physical units to ensure consistency with evaluation metrics. This normalization scheme improves model stability and comparability across heterogeneous variables.

Table 4: The data details.

| TASK | VARIABLE NAME | LAYERS | SPATIAL RESOLUTION | TEMPORAL FREQUENCY | LAT-LON RANGE |
|---|---|---|---|---|---|
| GLOBAL | GEOPOTENTIAL (Z) | 13 | $5.625°$ | 6H | $-90°$S–$90°$N, $180°$W–$180°$E |
| | SPECIFIC HUMIDITY (Q) | 13 | $5.625°$ | 6H | $-90°$S–$90°$N, $180°$W–$180°$E |
| | TEMPERATURE (T) | 13 | $5.625°$ | 6H | $-90°$S–$90°$N, $180°$W–$180°$E |
| | U COMPONENT OF WIND (U) | 13 | $5.625°$ | 6H | $-90°$S–$90°$N, $180°$W–$180°$E |
| | V COMPONENT OF WIND (V) | 13 | $5.625°$ | 6H | $-90°$S–$90°$N, $180°$W–$180°$E |
| | 10M U WIND (U10) | 1 | $5.625°$ | 6H | $-90°$S–$90°$N, $180°$W–$180°$E |
| | 10M V WIND (V10) | 1 | $5.625°$ | 6H | $-90°$S–$90°$N, $180°$W–$180°$E |
| | 2M TEMPERATURE (T2M) | 1 | $5.625°$ | 6H | $-90°$S–$90°$N, $180°$W–$180°$E |
| | MEAN SEA LEVEL PRESSURE (MSLP) | 1 | $5.625°$ | 6H | $-90°$S–$90°$N, $180°$W–$180°$E |
| | TOTAL PRECIPITATION (PREC) | 1 | $5.625°$ | 6H | $-90°$S–$90°$N, $180°$W–$180°$E |
| CONTINENTAL | GEOPOTENTIAL (Z) | 13 | $1.0°$ | 6H | $56°$S–$14°$N, $81°$W–$34°$W |
| | SPECIFIC HUMIDITY (Q) | 13 | $1.0°$ | 6H | SAME AS ABOVE |
| | TEMPERATURE (T) | 13 | $1.0°$ | 6H | SAME AS ABOVE |
| | U COMPONENT OF WIND (U) | 13 | $1.0°$ | 6H | SAME AS ABOVE |
| | V COMPONENT OF WIND (V) | 13 | $1.0°$ | 6H | SAME AS ABOVE |
| | 10M U WIND (U10) | 1 | $1.0°$ | 6H | SAME AS ABOVE |
| | 10M V WIND (V10) | 1 | $1.0°$ | 6H | SAME AS ABOVE |
| | 2M TEMPERATURE (T2M) | 1 | $1.0°$ | 6H | SAME AS ABOVE |
| | MEAN SEA LEVEL PRESSURE (MSLP) | 1 | $1.0°$ | 6H | SAME AS ABOVE |
| | TOTAL PRECIPITATION (PREC) | 1 | $1.0°$ | 6H | SAME AS ABOVE |
| REGIONAL | 2M TEMPERATURE (T2M) | 1 | $0.25°$ | 6H | $20°$N–$28°$N, $110°$E–$126°$E |
| | 10M U WIND (U10) | 1 | $0.25°$ | 6H | SAME AS ABOVE |
| | 10M V WIND (V10) | 1 | $0.25°$ | 6H | SAME AS ABOVE |

# D    MORE METRIC DETAILS

To evaluate forecasting skill, we follow standard practices in numerical weather prediction and report latitude-weighted Root Mean Squared Error (RMSE) and Anomaly Correlation Coefficient (ACC). Before computing the metrics, model outputs are de-normalized back to physical units for consistency with observations.

$$\text{RMSE} = \frac{1}{L} \sum_{\ell=1}^{L} \sqrt{\frac{1}{HW} \sum_{h=1}^{H} \sum_{w=1}^{W} \alpha(h) \, (y_{\ell h w} - \hat{x}_{\ell h w})^2}, \tag{11}$$

$$\text{ACC} = \frac{\sum_{\ell,h,w} \alpha(h) \, \tilde{y}_{\ell h w} \, \tilde{\hat{x}}_{\ell h w}}{\sqrt{\sum_{\ell,h,w} \alpha(h) \, \tilde{y}_{\ell h w}^2} \, \sqrt{\sum_{\ell,h,w} \alpha(h) \, \tilde{\hat{x}}_{\ell h w}^2}}, \tag{12}$$

where $\alpha(h) = \cos(h) \big/ \left( \frac{1}{H} \sum_{h'=1}^{H} \cos(h') \right)$ compensates for unequal grid areas across latitudes. The anomaly terms are defined as $\tilde{y}_{\ell h w} = y_{\ell h w} - C$ and $\tilde{\hat{x}}_{\ell h w} = \hat{x}_{\ell h w} - C$, with $C = \frac{1}{LHW} \sum_{\ell,h,w} y_{\ell h w}$ representing the climatological mean.

RMSE reflects the average magnitude of prediction errors while accounting for the Earth's geometry. ACC instead measures the similarity between predicted and observed anomalies, emphasizing the model's ability to capture dynamical patterns rather than absolute values. Together, these two metrics provide a balanced view of both error magnitude and anomaly-tracking skill.

# E    MORE IMPLEMENTATION DETAILS

All experiments are conducted on a server equipped with 8 NVIDIA A100 GPUs . Our implementation of **STORM** is based on PyTorch 2.1.0 (Paszke et al., 2019). For optimization, we adopt the Adam optimizer (Kingma, 2014) with a learning rate of $1 \times 10^{-3}$ and mean squared error (L2) as the training objective. Each model is trained for 100 epochs with early stopping based on the validation loss. The multi-scale hierarchy is constructed with $M = 3$ levels, and the hidden dimension is fixed to $D = 256$ across all experiments. The *Scale-Bridging Spatio-Temporal Encoder* is composed of $N = 3$ stacked layers. To ensure fairness, all baselines are re-trained under the same data preprocessing, optimization, and training protocols. The model is trained using the mean squared error (MSE) loss between predicted and observed atmospheric states, which aligns with the RMSE evaluation metric and encourages accurate recovery of both large-scale patterns and fine-grained variations.

# F    EFFICIENCY ANALYSIS

As shown in Table 5, STORM delivers a substantially better efficiency–accuracy trade-off compared with existing data-driven weather models. Despite having only 15.8M parameters, STORM achieves the lowest inference latency (0.87s per 100 samples) and competitive FLOPs, while also attaining the highest prediction accuracy (ACC = 0.984). In contrast, large-scale models such as Pangu and Fuxi incur significantly higher computational and memory costs but still fall short in accuracy. These results highlight that STORM's lightweight architecture effectively preserves predictive skill while enabling fast and resource-efficient forecasting.

Table 5: Efficiency comparison across representative data-driven weather models.

| Metric | STORM | Pangu | FCN | Fuxi |
|---|---|---|---|---|
| **Inference Time** ↓ (Seconds / 100 Samples) | **0.87** | 5.11 | 6.99 | 19.37 |
| **Peak GPU Memory** ↓ (MB, Batch Size = 1) | **729.22** | 408.75 | 265.95 | 2533.25 |
| **Parameters** ↓ (M) | **15.77** | 97.48 | 64.65 | 661.01 |
| **FLOPs** ↓ (GFLOPs) | **84.59** | 91.26 | 58.50 | 302.46 |
| **ACC** ↑ | **0.984** | 0.952 | 0.933 | 0.950 |

# G    FULL SENSITIVE ANALYSIS

Table 5 summarizes the scaling behavior of STORM under varying branch numbers and model widths. Increasing either the branch depth or $d_{\text{model}}$ consistently improves accuracy, with ACC rising from 0.941 (1 branch, 64 hidden units) to 0.988 (5 branches, 256–512 units), while RMSE steadily decreases. Notably, these gains come with only moderate growth in FLOPs and inference time, indicating that STORM's multi-branch temporal encoder scales efficiently. The results highlight a clear efficiency–accuracy trend: larger branches yield stronger predictive skill without incurring prohibitive computational cost.

Table 6: Scaling and efficiency analysis of STORM across different branch configurations.

| Branch | d_model | Total Params | Temp. Enc. Params | FLOPs | Time | Max Mem. | ACC | RMSE |
|---|---|---|---|---|---|---|---|---|
| 1 | 64 | 0.32 | 2.96E-04 | 3.17 | 0.28 | 13.75 | 0.941 | 16.415 |
| 1 | 128 | 1.20 | 2.96E-04 | 11.04 | 0.30 | 17.44 | 0.951 | 16.323 |
| 1 | 256 | 4.59 | 2.96E-04 | 40.88 | 0.30 | 30.01 | 0.954 | 16.267 |
| 1 | 512 | 17.96 | 2.96E-04 | 156.92 | 0.30 | 82.59 | 0.958 | 16.177 |
| 2 | 64 | 0.64 | 5.92E-04 | 4.46 | 0.56 | 14.97 | 0.963 | 16.136 |
| 2 | 128 | 2.45 | 5.92E-04 | 16.00 | 0.56 | 24.43 | 0.966 | 16.115 |
| 2 | 256 | 9.59 | 5.92E-04 | 60.32 | 0.56 | 51.66 | 0.971 | 16.093 |
| 2 | 512 | 37.92 | 5.92E-04 | 233.92 | 0.59 | 164.49 | 0.974 | 15.983 |
| 3 | 64 | 1.03 | 8.88E-04 | 6.05 | 0.86 | 19.48 | 0.977 | 15.913 |
| 3 | 128 | 4.00 | 8.88E-04 | 22.16 | 0.85 | 30.35 | 0.981 | 15.854 |
| 3 | 256 | 15.77 | 8.88E-04 | 84.59 | 0.87 | 75.48 | 0.984 | 15.809 |
| 3 | 512 | 62.60 | 8.88E-04 | 330.25 | 0.87 | 257.33 | 0.985 | 15.798 |
| 4 | 64 | 1.46 | 1.18E-03 | 7.44 | 1.15 | 20.31 | 0.979 | 15.841 |
| 4 | 128 | 5.70 | 1.18E-03 | 27.61 | 1.16 | 37.27 | 0.983 | 15.813 |
| 4 | 256 | 22.54 | 1.18E-03 | 106.12 | 1.17 | 101.06 | 0.987 | 15.769 |
| 4 | 512 | 89.64 | 1.18E-03 | 415.84 | 1.20 | 360.60 | 0.986 | 15.784 |
| 5 | 64 | 1.96 | 1.48E-03 | 8.23 | 1.49 | 335.52 | 0.983 | 15.808 |
| 5 | 128 | 7.69 | 1.48E-03 | 30.62 | 1.52 | 361.66 | 0.984 | 15.796 |
| 5 | 256 | 30.49 | 1.48E-03 | 117.84 | 1.51 | 448.31 | 0.988 | 15.757 |
| 5 | 512 | 121.40 | 1.48E-03 | 462.11 | 1.53 | 908.50 | 0.988 | 15.716 |

# H    MORE DETAILS OF ABLATION ANALYSIS

Tables 7 and 8 highlight the contribution of message passing and scale heterogeneity in STORM's multi-branch architecture. First, enabling message passing consistently boosts performance across all branch counts, with ACC improvements of 0.6–1.0% and reduced RMSE, indicating more effective cross-scale information flow. Second, branches configured with different temporal scales significantly outperform those using identical scales, showing clear gains that grow with larger branch numbers. These results confirm that both cross-scale communication and scale diversity are essential for extracting complementary temporal patterns and achieving stronger predictive skill.

Table 7: Ablation on message passing across multi-scale branches.

| Branch (Scale Numbers) | ACC (with MP) | RMSE (with MP) | ACC (without MP) | RMSE (without MP) |
|---|---|---|---|---|
| 2 | 0.971 | 16.093 | 0.963 | 16.145 |
| 3 | 0.984 | 15.809 | 0.976 | 15.948 |
| 4 | 0.987 | 15.769 | 0.981 | 15.886 |
| 5 | 0.988 | 15.757 | 0.980 | 16.879 |

Table 8: Effect of different vs same multi-branch designs.

| Scale Number | ACC (Diff Scale) | RMSE (Diff) | ACC (Same Scale) | RMSE (Same) |
|---|---|---|---|---|
| 2 | 0.971 | 16.093 | 0.939 | 16.547 |
| 3 | 0.984 | 15.809 | 0.948 | 16.382 |
| 4 | 0.987 | 15.769 | 0.954 | 16.285 |
| 5 | 0.988 | 15.757 | 0.957 | 16.179 |

# I FORECAST VISUALIZATION

## I.1 MEAN SEA LEVEL PRESSURE (MSLP)

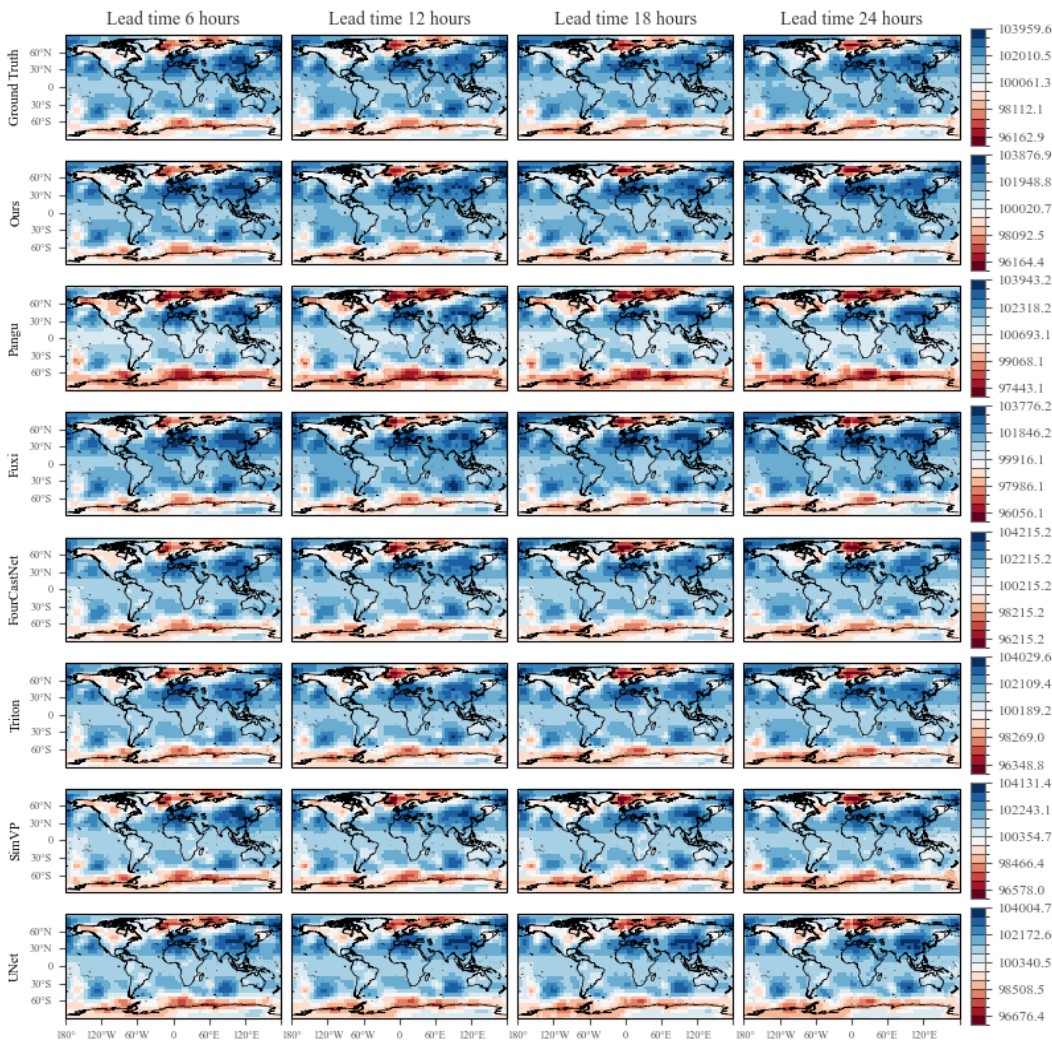

Figure 8: 24-hour forecast results of different models for mean sea level pressure (MSLP).

## I.2 SPECIFIC HUMIDITY AT 500 HPA (Q500)

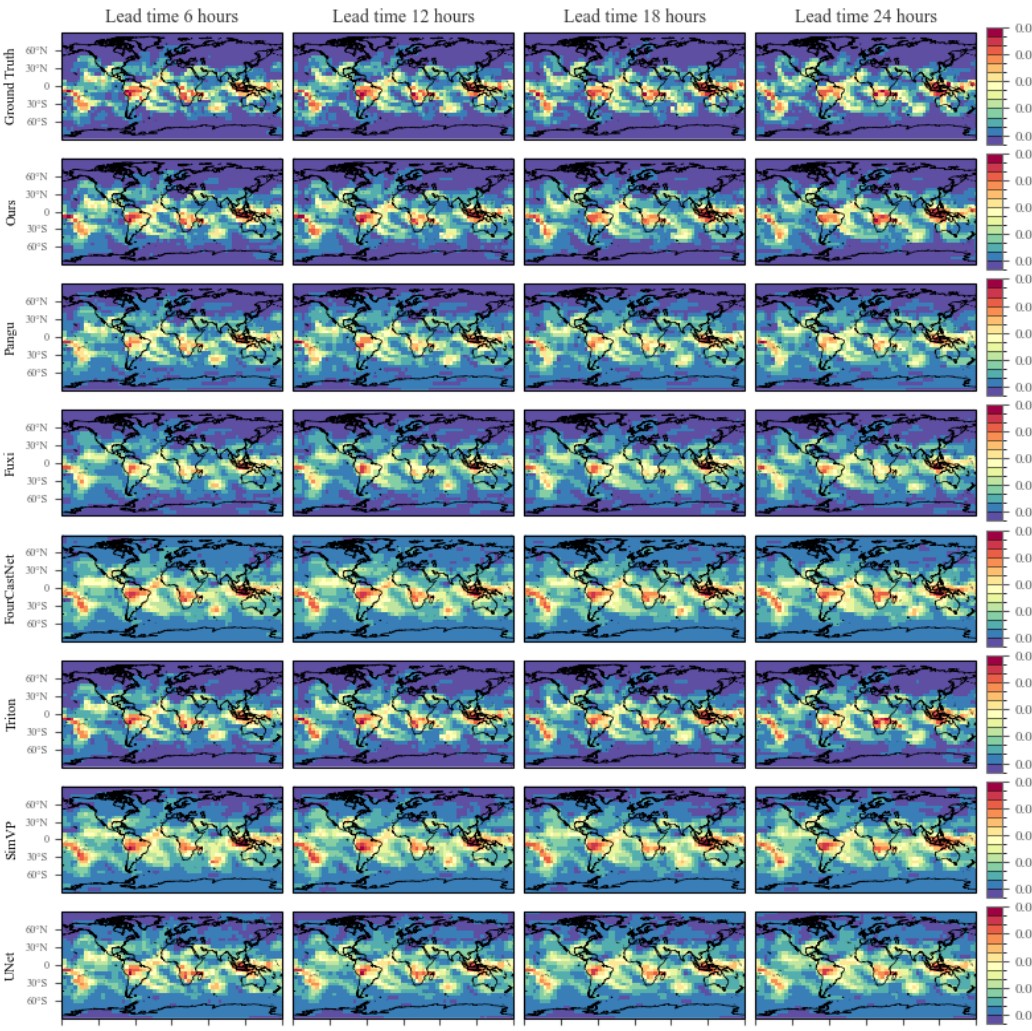

Figure 9: 24-hour forecast results of different models for 500 hPa specific humidity (Q500).

## I.3 PRECIPITATION

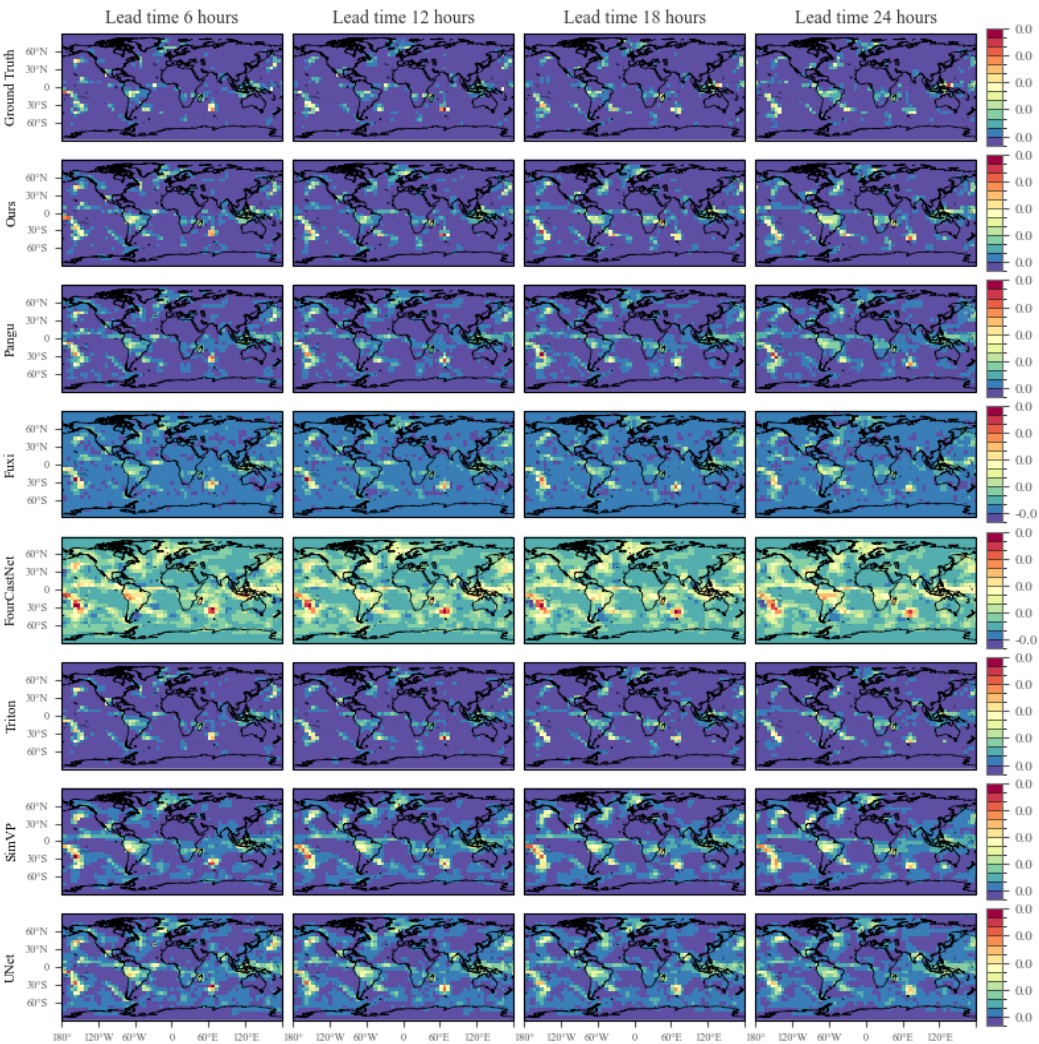

Figure 10: 24-hour precipitation forecast results of different models.

## I.4 2-METER TEMPERATURE (T2M)

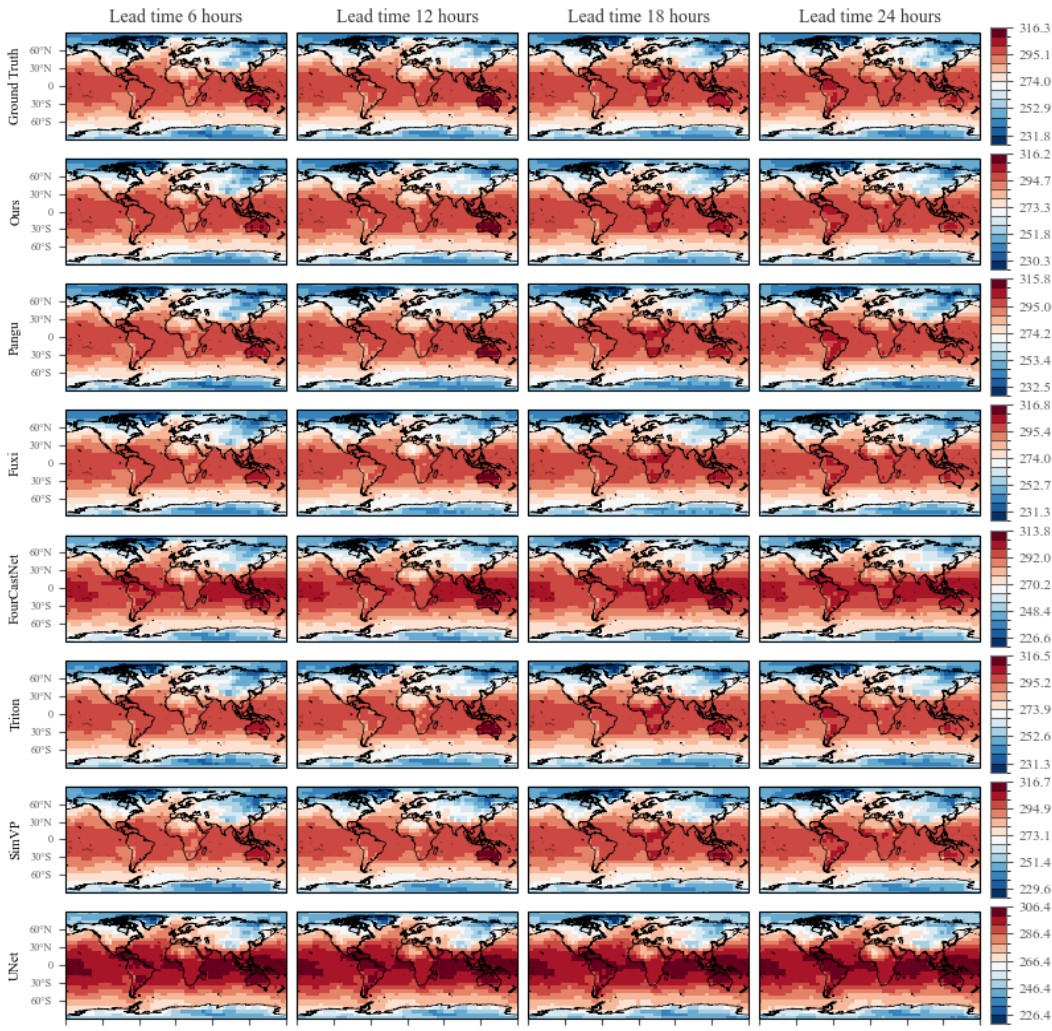

Figure 11: 24-hour forecast results of different models for 2-meter temperature (T2M).

## I.5 TEMPERATURE AT 500 HPA (T500)

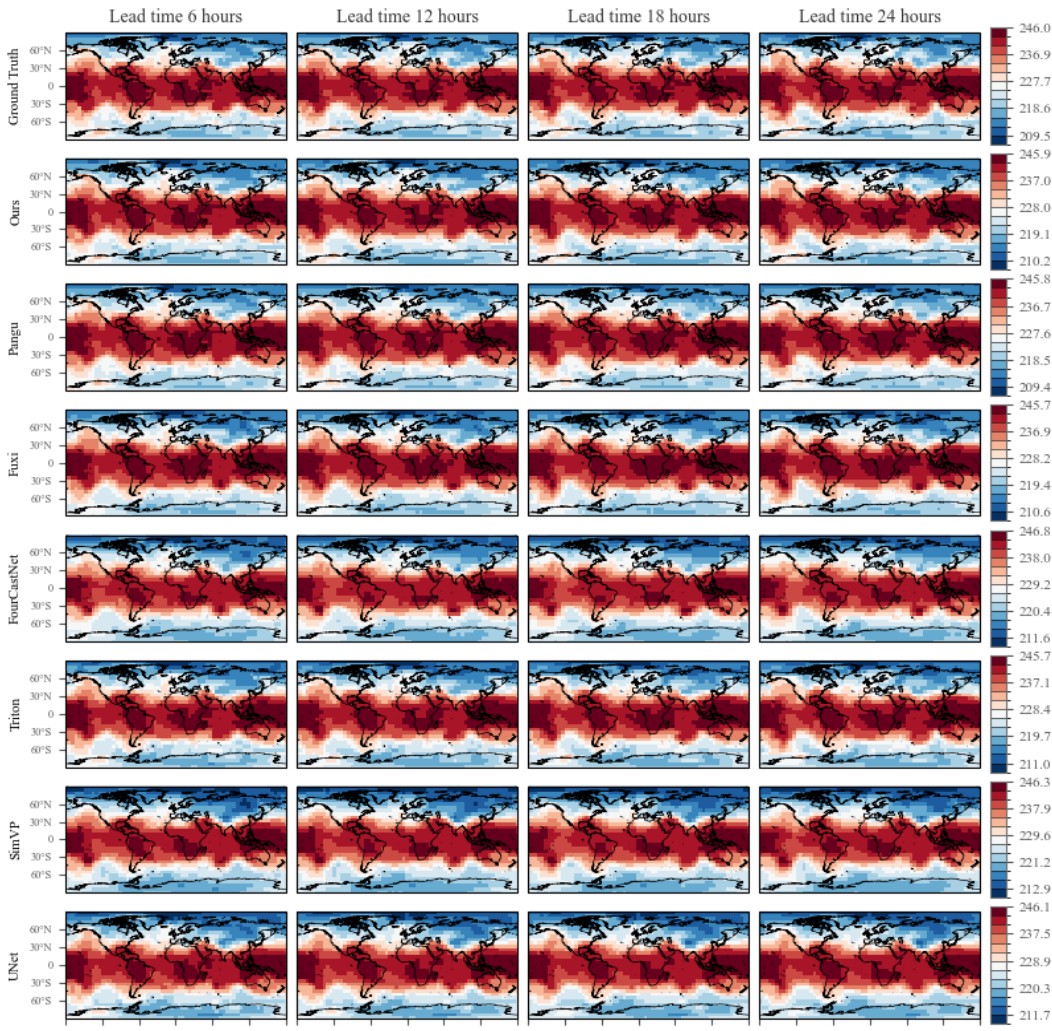

Figure 12: 24-hour forecast results of different models for 500 hPa temperature (T500).

## I.6    10-METER ZONAL WIND (U10)

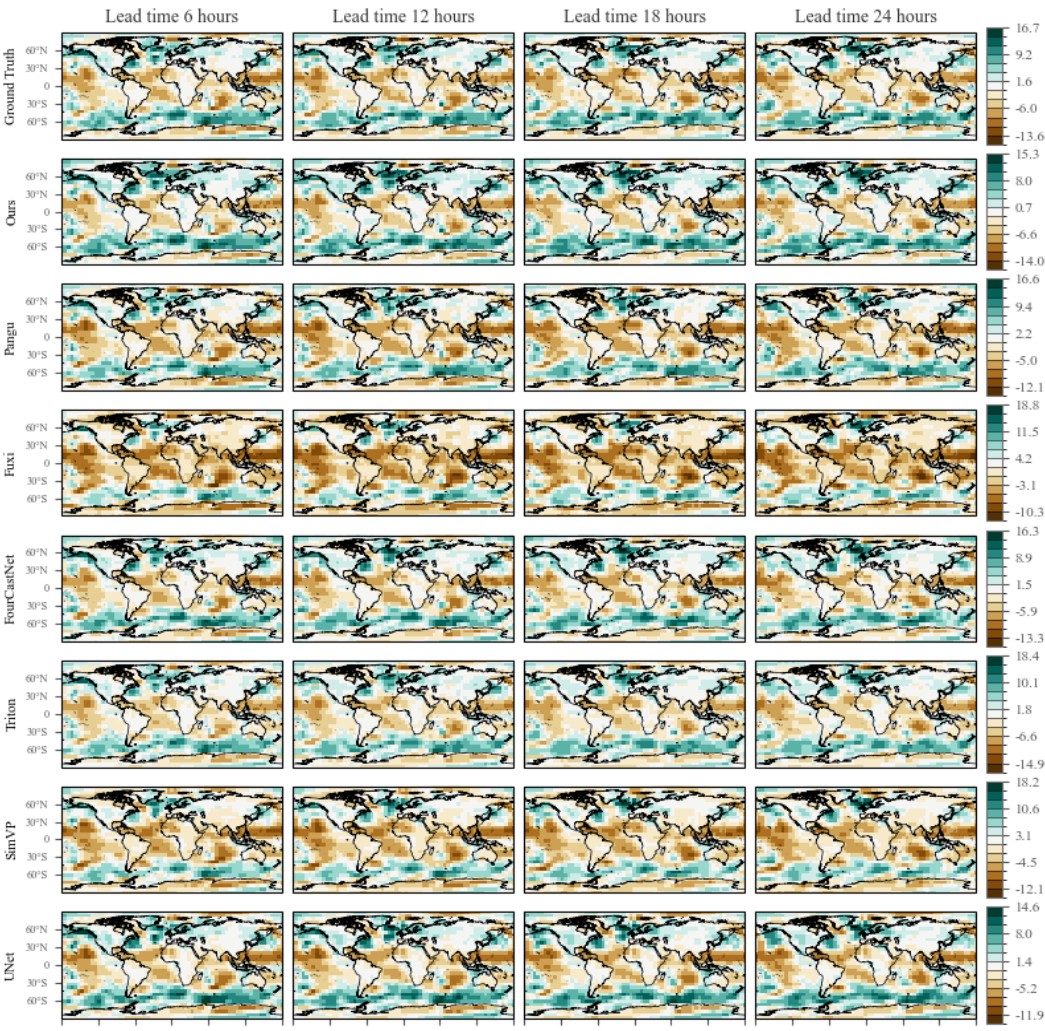

Figure 13: 24-hour forecast results of different models for 10-meter zonal wind (U10).

## I.7 ZONAL WIND AT 500 HPA (U500)

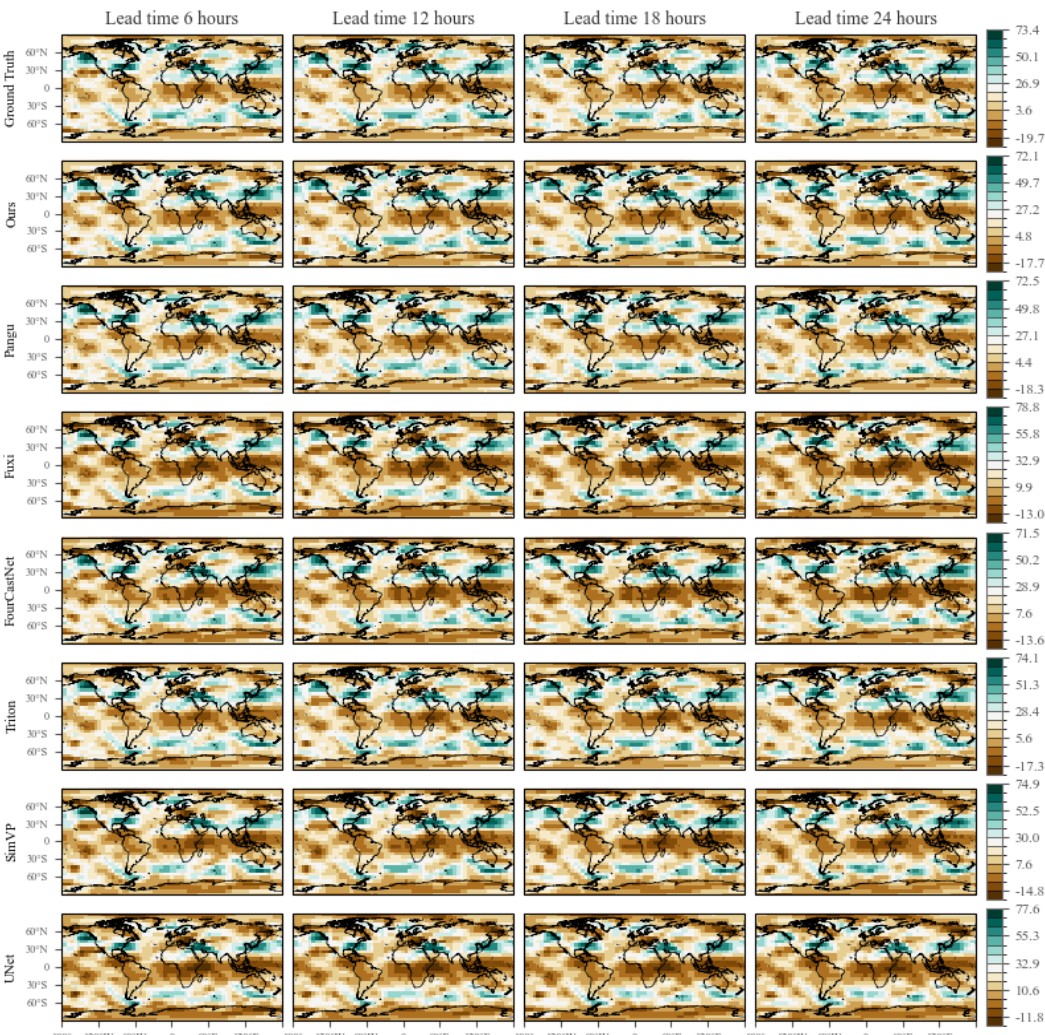

Figure 14: 24-hour forecast results of different models for 500 hPa zonal wind (U500).

## I.8  10-METER MERIDIONAL WIND (V10)

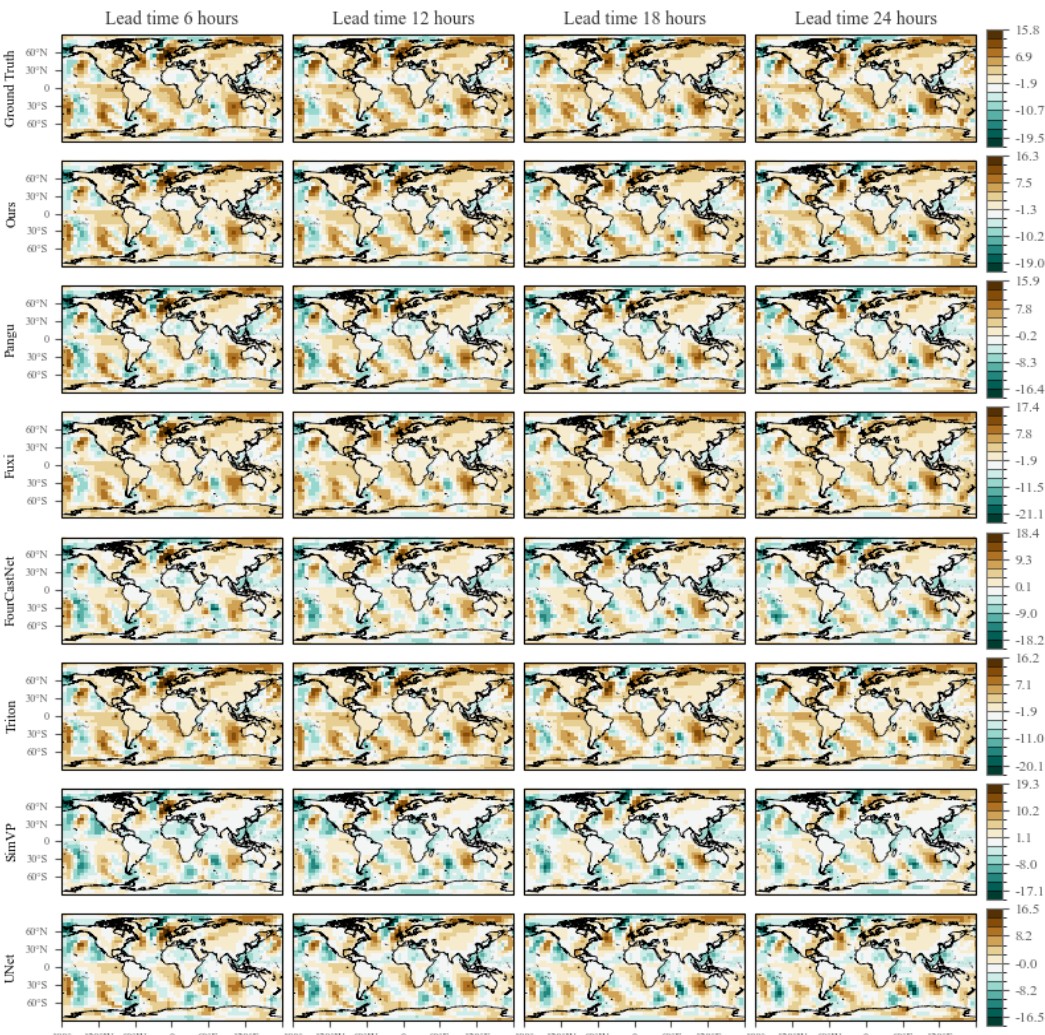

Figure 15: 24-hour forecast results of different models for 10-meter meridional wind (V10).

## I.9  MERIDIONAL WIND AT 500 HPA (V500)

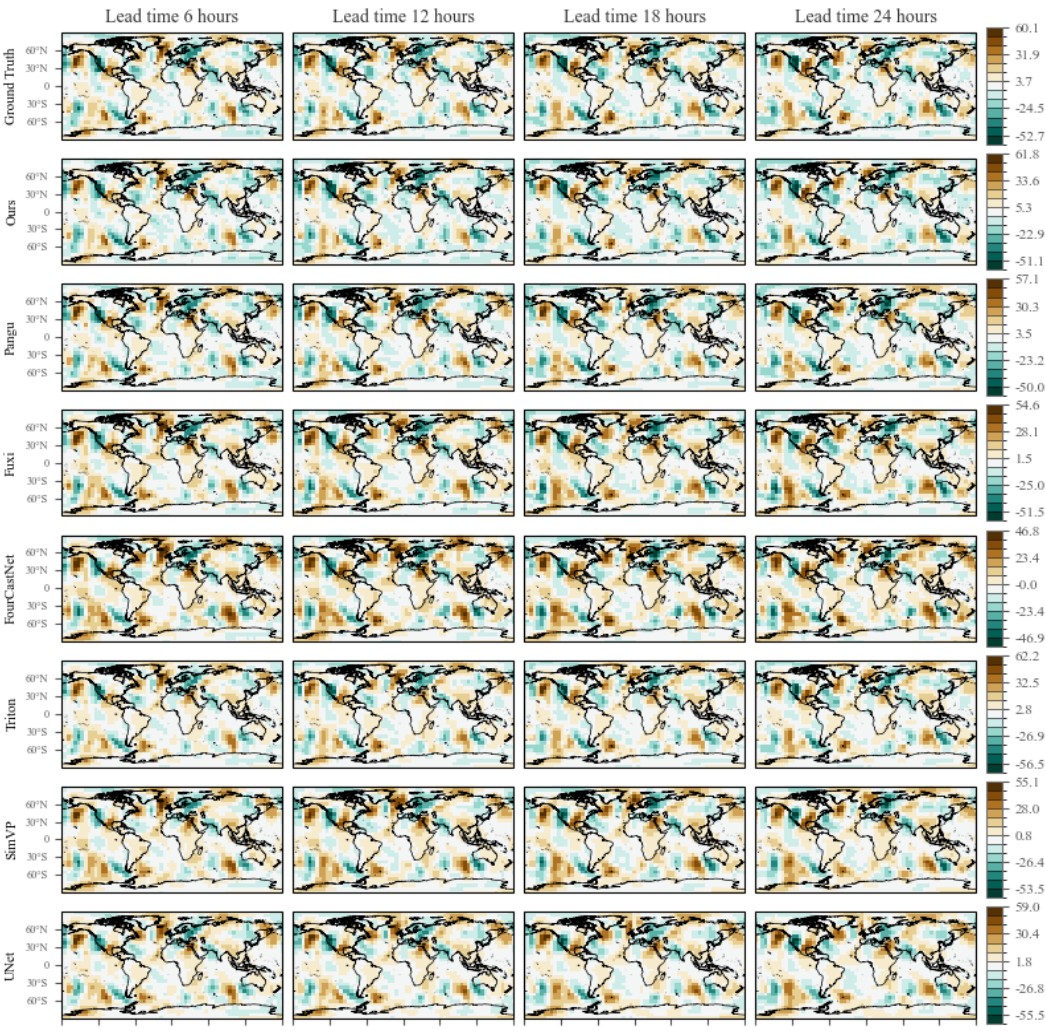

Figure 16: 24-hour forecast results of different models for 500 hPa meridional wind (V500).

## I.10 GEOPOTENTIAL HEIGHT AT 500 HPA (Z500)

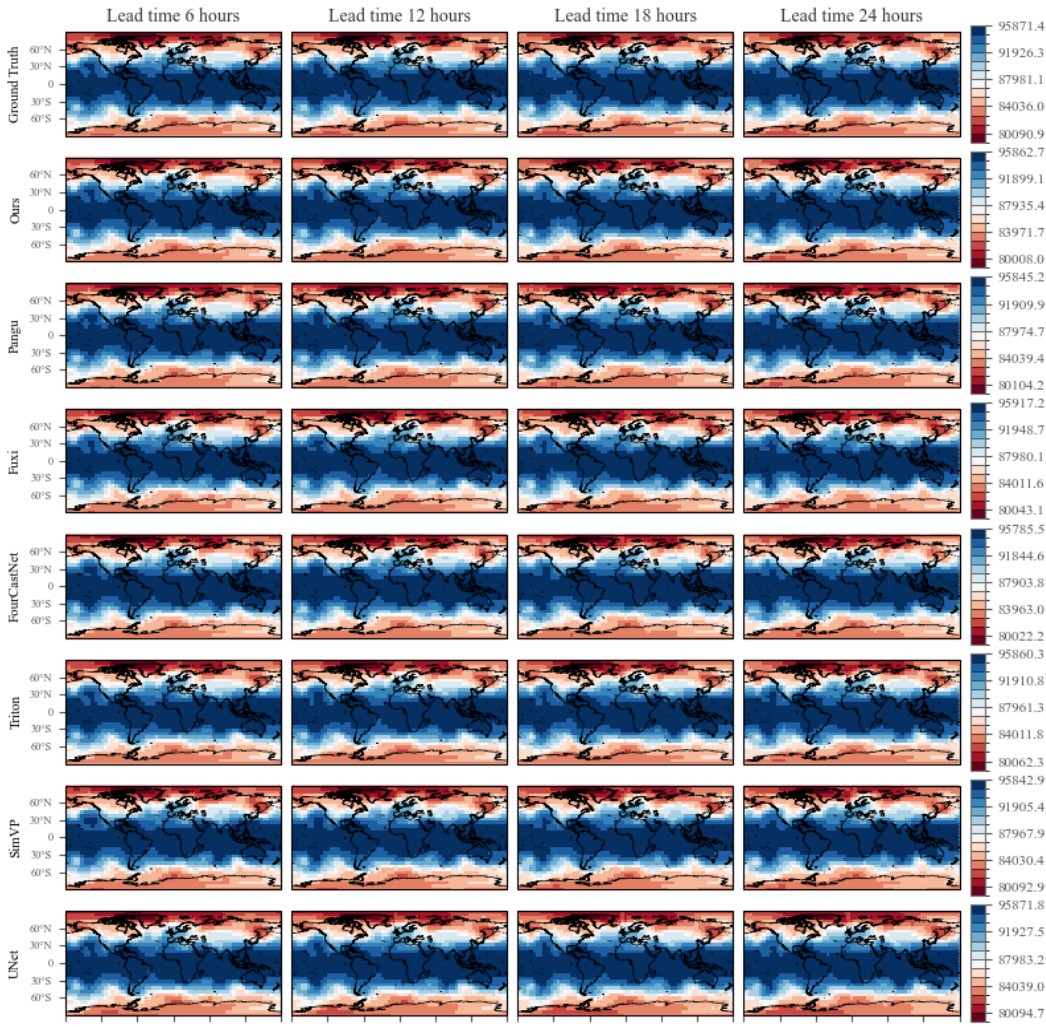

Figure 17: 24-hour forecast results of different models for 500 hPa geopotential height (Z500).

## J   FULL SHORT-TERM FORECASTING RESULTS.

.

Table 9: Quantitative comparison of short-term (up to 24 hours) global weather forecasting performance.

| Metric | | RMSE | | | | | | | ACC | | | | | | |
|---|---|---|---|---|---|---|---|---|---|---|---|---|---|---|---|
| Variable | Hours | Ours | Triton | Pangu | FCN | Fuxi | SimVP | UNet | Ours | Triton | Pangu | FCN | Fuxi | SimVP | UNet |
| T2M | 6h | 0.564 | 0.718 | 0.787 | 1.430 | 1.191 | 1.335 | 1.337 | 0.999 | 0.999 | 0.998 | 0.995 | 0.996 | 0.996 | 0.995 |
| | 12h | 0.653 | 0.836 | 1.075 | 1.545 | 1.308 | 1.332 | 2.208 | 0.999 | 0.998 | 0.997 | 0.994 | 0.996 | 0.996 | 0.987 |
| | 18h | 0.705 | 0.929 | 1.219 | 1.632 | 1.415 | 1.341 | 2.789 | 0.999 | 0.998 | 0.996 | 0.993 | 0.995 | 0.996 | 0.982 |
| | 24h | 0.777 | 1.010 | 1.342 | 1.708 | 1.513 | 1.363 | 3.447 | 0.998 | 0.997 | 0.995 | 0.993 | 0.994 | 0.995 | 0.981 |
| U10 | 6h | 0.495 | 0.598 | 0.775 | 1.169 | 0.994 | 1.279 | 0.955 | 0.995 | 0.993 | 0.988 | 0.973 | 0.981 | 0.968 | 0.981 |
| | 12h | 0.608 | 0.749 | 1.209 | 1.376 | 1.200 | 1.359 | 1.437 | 0.993 | 0.989 | 0.972 | 0.963 | 0.972 | 0.964 | 0.961 |
| | 18h | 0.719 | 0.894 | 1.590 | 1.646 | 1.423 | 1.497 | 2.032 | 0.990 | 0.985 | 0.951 | 0.947 | 0.961 | 0.956 | 0.942 |
| | 24h | 0.853 | 1.034 | 1.933 | 1.926 | 1.637 | 1.659 | 3.462 | 0.986 | 0.979 | 0.927 | 0.927 | 0.948 | 0.946 | 0.924 |
| V10 | 6h | 0.520 | 0.629 | 0.834 | 1.274 | 1.064 | 1.378 | 1.012 | 0.992 | 0.988 | 0.979 | 0.950 | 0.965 | 0.941 | 0.967 |
| | 12h | 0.647 | 0.796 | 1.304 | 1.492 | 1.268 | 1.467 | 1.491 | 0.987 | 0.981 | 0.947 | 0.930 | 0.950 | 0.933 | 0.933 |
| | 18h | 0.768 | 0.950 | 1.713 | 1.782 | 1.490 | 1.608 | 1.994 | 0.982 | 0.973 | 0.908 | 0.899 | 0.932 | 0.919 | 0.898 |
| | 24h | 0.917 | 1.098 | 2.082 | 2.084 | 1.703 | 1.772 | 2.971 | 0.974 | 0.963 | 0.862 | 0.861 | 0.911 | 0.900 | 0.863 |
| Prec | 6h | 4.2E-04 | 4.8E-04 | 6.1E-04 | 9.2E-04 | 7.4E-04 | 9.2E-04 | 6.7E-04 | 0.955 | 0.941 | 0.906 | 0.763 | 0.856 | 0.768 | 0.885 |
| | 12h | 4.9E-04 | 5.8E-04 | 7.9E-04 | 9.5E-04 | 8.2E-04 | 9.3E-04 | 8.9E-04 | 0.938 | 0.913 | 0.833 | 0.748 | 0.819 | 0.758 | 0.792 |
| | 18h | 5.9E-04 | 7.0E-04 | 9.3E-04 | 1.0E-03 | 9.2E-04 | 9.7E-04 | 1.1E-03 | 0.911 | 0.872 | 0.765 | 0.717 | 0.768 | 0.737 | 0.719 |
| | 24h | 6.6E-04 | 7.8E-04 | 1.0E-03 | 1.1E-03 | 1.0E-03 | 1.0E-03 | 1.4E-03 | 0.886 | 0.839 | 0.705 | 0.682 | 0.722 | 0.713 | 0.663 |
| MSLP | 6h | 46.808 | 60.493 | 101.678 | 147.757 | 112.764 | 154.169 | 113.815 | 0.999 | 0.999 | 0.996 | 0.991 | 0.995 | 0.990 | 0.994 |
| | 12h | 61.834 | 82.905 | 179.577 | 186.644 | 148.871 | 168.679 | 198.949 | 0.998 | 0.997 | 0.987 | 0.986 | 0.991 | 0.989 | 0.984 |
| | 18h | 78.448 | 104.797 | 255.410 | 238.317 | 188.986 | 194.126 | 289.322 | 0.998 | 0.996 | 0.973 | 0.977 | 0.986 | 0.985 | 0.971 |
| | 24h | 99.948 | 126.514 | 326.712 | 295.758 | 229.717 | 226.092 | 427.862 | 0.996 | 0.994 | 0.956 | 0.964 | 0.979 | 0.979 | 0.956 |
| U500 | 6h | 1.384 | 1.835 | 1.879 | 3.540 | 2.953 | 3.553 | 2.503 | 0.996 | 0.994 | 0.993 | 0.976 | 0.983 | 0.976 | 0.987 |
| | 12h | 1.713 | 2.287 | 2.972 | 4.148 | 3.612 | 3.789 | 3.880 | 0.994 | 0.990 | 0.983 | 0.967 | 0.975 | 0.973 | 0.971 |
| | 18h | 2.041 | 2.725 | 3.945 | 4.940 | 4.303 | 4.204 | 5.158 | 0.992 | 0.986 | 0.970 | 0.953 | 0.965 | 0.966 | 0.955 |
| | 24h | 2.472 | 3.151 | 4.845 | 5.802 | 4.980 | 4.719 | 7.166 | 0.988 | 0.981 | 0.955 | 0.935 | 0.953 | 0.957 | 0.939 |
| V500 | 6h | 1.449 | 1.873 | 2.036 | 3.765 | 3.155 | 3.823 | 2.868 | 0.992 | 0.987 | 0.985 | 0.947 | 0.963 | 0.945 | 0.969 |
| | 12h | 1.835 | 2.367 | 3.228 | 4.486 | 3.817 | 4.093 | 4.259 | 0.987 | 0.979 | 0.961 | 0.924 | 0.946 | 0.936 | 0.933 |
| | 18h | 2.221 | 2.843 | 4.290 | 5.418 | 4.500 | 4.530 | 5.508 | 0.982 | 0.970 | 0.930 | 0.887 | 0.925 | 0.922 | 0.897 |
| | 24h | 2.714 | 3.305 | 5.284 | 6.414 | 5.169 | 5.073 | 7.363 | 0.972 | 0.960 | 0.893 | 0.841 | 0.902 | 0.901 | 0.859 |
| T500 | 6h | 0.376 | 0.518 | 0.513 | 1.087 | 0.903 | 1.189 | 0.679 | 0.999 | 0.998 | 0.998 | 0.993 | 0.995 | 0.992 | 0.997 |
| | 12h | 0.478 | 0.646 | 0.802 | 1.177 | 1.036 | 1.210 | 1.087 | 0.999 | 0.998 | 0.996 | 0.992 | 0.994 | 0.991 | 0.993 |
| | 18h | 0.566 | 0.760 | 1.024 | 1.300 | 1.182 | 1.255 | 1.410 | 0.998 | 0.997 | 0.994 | 0.990 | 0.992 | 0.991 | 0.988 |
| | 24h | 0.675 | 0.861 | 1.207 | 1.441 | 1.321 | 1.313 | 1.773 | 0.997 | 0.996 | 0.991 | 0.988 | 0.990 | 0.990 | 0.985 |
| Z500 | 6h | 51.5 | 68.4 | 102.6 | 189.0 | 142.3 | 186.4 | 131.0 | 1.000 | 1.000 | 1.000 | 0.999 | 0.999 | 0.999 | 0.999 |
| | 12h | 66.3 | 93.4 | 184.6 | 223.6 | 185.5 | 200.9 | 233.4 | 1.000 | 1.000 | 0.999 | 0.999 | 0.999 | 0.999 | 0.998 |
| | 18h | 86.5 | 122.5 | 267.1 | 289.9 | 238.5 | 231.9 | 348.0 | 1.000 | 1.000 | 0.998 | 0.997 | 0.998 | 0.998 | 0.996 |
| | 24h | 114.9 | 153.3 | 348.5 | 371.5 | 294.8 | 274.6 | 520.8 | 1.000 | 0.999 | 0.996 | 0.996 | 0.997 | 0.998 | 0.995 |
| Q500 | 6h | 2.8E-05 | 3.7E-05 | 3.4E-05 | 6.5E-05 | 5.8E-05 | 7.5E-05 | 4.3E-05 | 0.988 | 0.979 | 0.983 | 0.936 | 0.949 | 0.914 | 0.972 |
| | 12h | 3.7E-05 | 4.6E-05 | 5.2E-05 | 7.0E-05 | 6.5E-05 | 7.5E-05 | 6.5E-05 | 0.980 | 0.968 | 0.960 | 0.926 | 0.935 | 0.913 | 0.937 |
| | 18h | 4.3E-05 | 5.4E-05 | 6.4E-05 | 7.5E-05 | 7.3E-05 | 7.7E-05 | 8.1E-05 | 0.972 | 0.956 | 0.937 | 0.912 | 0.918 | 0.908 | 0.906 |
| | 24h | 4.9E-05 | 6.0E-05 | 7.4E-05 | 8.1E-05 | 8.0E-05 | 7.9E-05 | 1.0E-04 | 0.963 | 0.945 | 0.916 | 0.897 | 0.900 | 0.902 | 0.881 |

## K  FULL LONG-TERM FORECASTING RESULTS.

.

Table 10: Quantitative comparison of long-term (7–10 days) global weather forecasting performance.

| Metric | | RMSE | | | | | | | ACC | | | | | | |
|---|---|---|---|---|---|---|---|---|---|---|---|---|---|---|---|
| Variable | Days | Ours | Triton | Pangu | FCN | Fuxi | SimVP | UNet | Ours | Triton | Pangu | FCN | Fuxi | SimVP | UNet |
| T2M | 7d | 2.293 | 2.934 | 3.423 | 3.902 | 3.294 | 2.938 | 4.051 | 0.987 | 0.974 | 0.963 | 0.975 | 0.973 | 0.979 | 0.948 |
| | 8d | 2.516 | 3.263 | 3.574 | 4.104 | 3.442 | 3.100 | 4.358 | 0.985 | 0.968 | 0.958 | 0.971 | 0.970 | 0.977 | 0.940 |
| | 9d | 2.708 | 3.549 | 3.712 | 4.288 | 3.574 | 3.248 | 4.630 | 0.982 | 0.961 | 0.952 | 0.969 | 0.968 | 0.974 | 0.933 |
| | 10d | 2.865 | 3.799 | 3.821 | 4.430 | 3.689 | 3.384 | 4.867 | 0.981 | 0.956 | 0.948 | 0.967 | 0.966 | 0.972 | 0.928 |
| U10 | 7d | 3.485 | 3.945 | 4.959 | 5.102 | 4.528 | 3.766 | 5.507 | 0.769 | 0.700 | 0.663 | 0.645 | 0.608 | 0.707 | 0.606 |
| | 8d | 3.773 | 4.192 | 5.088 | 5.301 | 4.660 | 3.872 | 5.753 | 0.730 | 0.662 | 0.627 | 0.614 | 0.582 | 0.689 | 0.567 |
| | 9d | 3.954 | 4.397 | 5.192 | 5.464 | 4.762 | 3.997 | 5.953 | 0.697 | 0.631 | 0.597 | 0.588 | 0.561 | 0.675 | 0.541 |
| | 10d | 4.013 | 4.548 | 5.268 | 5.583 | 4.834 | 4.174 | 6.099 | 0.670 | 0.606 | 0.572 | 0.568 | 0.544 | 0.664 | 0.524 |
| V10 | 7d | 3.787 | 4.205 | 4.961 | 5.364 | 4.652 | 3.982 | 5.757 | 0.559 | 0.451 | 0.392 | 0.358 | 0.326 | 0.427 | 0.306 |
| | 8d | 4.089 | 4.445 | 5.069 | 5.514 | 4.764 | 4.098 | 5.986 | 0.486 | 0.388 | 0.342 | 0.310 | 0.291 | 0.391 | 0.271 |
| | 9d | 4.154 | 4.620 | 5.150 | 5.649 | 4.853 | 4.335 | 6.158 | 0.426 | 0.339 | 0.299 | 0.274 | 0.264 | 0.366 | 0.244 |
| | 10d | 4.201 | 4.747 | 5.201 | 5.749 | 4.911 | 4.500 | 6.277 | 0.383 | 0.302 | 0.266 | 0.247 | 0.245 | 0.349 | 0.225 |
| Prec | 7d | 1.3E-03 | 1.9E-03 | 1.7E-03 | 1.5E-03 | 1.6E-03 | 1.4E-03 | 2.2E-03 | 0.492 | 0.354 | 0.329 | 0.259 | 0.271 | 0.396 | 0.242 |
| | 8d | 1.4E-03 | 2.1E-03 | 1.8E-03 | 1.6E-03 | 1.6E-03 | 1.5E-03 | 2.3E-03 | 0.446 | 0.316 | 0.297 | 0.231 | 0.249 | 0.384 | 0.211 |
| | 9d | 1.4E-03 | 2.2E-03 | 1.8E-03 | 1.6E-03 | 1.7E-03 | 1.5E-03 | 2.3E-03 | 0.410 | 0.287 | 0.272 | 0.210 | 0.233 | 0.376 | 0.190 |
| | 10d | 1.4E-03 | 2.3E-03 | 1.8E-03 | 1.6E-03 | 1.7E-03 | 1.6E-03 | 2.4E-03 | 0.381 | 0.262 | 0.252 | 0.193 | 0.220 | 0.370 | 0.173 |
| MSLP | 7d | 640.6 | 731.4 | 1075.0 | 969.2 | 901.1 | 741.8 | 1067.9 | 0.835 | 0.779 | 0.785 | 0.764 | 0.680 | 0.770 | 0.733 |
| | 8d | 713.0 | 795.5 | 1114.5 | 1024.9 | 940.6 | 775.8 | 1128.5 | 0.796 | 0.738 | 0.754 | 0.730 | 0.650 | 0.748 | 0.695 |
| | 9d | 771.7 | 847.9 | 1145.0 | 1069.5 | 969.0 | 801.4 | 1181.0 | 0.762 | 0.706 | 0.725 | 0.701 | 0.627 | 0.730 | 0.660 |
| | 10d | 816.2 | 887.7 | 1166.9 | 1101.8 | 989.2 | 820.0 | 1220.0 | 0.734 | 0.678 | 0.700 | 0.678 | 0.608 | 0.716 | 0.630 |
| U925 | 7d | 5.661 | 7.696 | 10.090 | 8.300 | 9.408 | 7.572 | 11.118 | 0.915 | 0.839 | 0.842 | 0.833 | 0.782 | 0.848 | 0.796 |
| | 8d | 6.320 | 8.590 | 10.607 | 8.936 | 9.972 | 7.972 | 12.066 | 0.894 | 0.802 | 0.816 | 0.808 | 0.756 | 0.832 | 0.763 |
| | 9d | 6.911 | 9.354 | 11.050 | 9.477 | 10.426 | 8.313 | 12.659 | 0.874 | 0.769 | 0.789 | 0.779 | 0.734 | 0.819 | 0.730 |
| | 10d | 7.436 | 10.031 | 11.430 | 9.921 | 10.803 | 8.611 | 13.103 | 0.854 | 0.739 | 0.765 | 0.752 | 0.716 | 0.807 | 0.699 |
| V925 | 7d | 5.674 | 7.095 | 9.228 | 8.929 | 8.310 | 6.765 | 10.013 | 0.643 | 0.485 | 0.395 | 0.376 | 0.289 | 0.416 | 0.300 |
| | 8d | 6.344 | 7.733 | 9.602 | 9.466 | 8.704 | 7.061 | 10.684 | 0.556 | 0.397 | 0.323 | 0.307 | 0.226 | 0.355 | 0.223 |
| | 9d | 6.923 | 8.210 | 9.892 | 9.880 | 8.981 | 7.294 | 11.174 | 0.474 | 0.327 | 0.263 | 0.250 | 0.183 | 0.305 | 0.166 |
| | 10d | 7.412 | 8.603 | 10.105 | 10.198 | 9.182 | 7.473 | 11.560 | 0.402 | 0.268 | 0.214 | 0.204 | 0.151 | 0.266 | 0.131 |
| T925 | 7d | 2.420 | 3.251 | 4.015 | 3.496 | 4.163 | 3.333 | 4.872 | 0.975 | 0.952 | 0.944 | 0.949 | 0.925 | 0.952 | 0.920 |
| | 8d | 2.690 | 3.585 | 4.232 | 3.777 | 4.421 | 3.504 | 5.204 | 0.969 | 0.942 | 0.935 | 0.942 | 0.916 | 0.947 | 0.908 |
| | 9d | 2.932 | 3.882 | 4.420 | 4.022 | 4.642 | 3.649 | 5.523 | 0.963 | 0.932 | 0.926 | 0.934 | 0.908 | 0.943 | 0.896 |
| | 10d | 3.145 | 4.152 | 4.566 | 4.223 | 4.827 | 3.774 | 5.724 | 0.957 | 0.922 | 0.917 | 0.926 | 0.901 | 0.939 | 0.884 |
| Z925 | 7d | 3.0E-07 | 4.7E-07 | 4.2E-07 | 4.3E-07 | 5.2E-07 | 3.8E-07 | 5.8E-07 | 0.782 | 0.611 | 0.607 | 0.620 | 0.567 | 0.653 | 0.565 |
| | 8d | 3.2E-07 | 5.2E-07 | 4.4E-07 | 4.6E-07 | 5.5E-07 | 3.9E-07 | 6.3E-07 | 0.760 | 0.573 | 0.571 | 0.586 | 0.544 | 0.637 | 0.524 |
| | 9d | 3.3E-07 | 5.6E-07 | 4.6E-07 | 4.8E-07 | 5.7E-07 | 4.1E-07 | 6.6E-07 | 0.738 | 0.537 | 0.537 | 0.554 | 0.524 | 0.622 | 0.504 |
| | 10d | 3.5E-07 | 6.0E-07 | 4.8E-07 | 5.0E-07 | 5.8E-07 | 4.2E-07 | 6.8E-07 | 0.717 | 0.502 | 0.509 | 0.527 | 0.506 | 0.609 | 0.482 |
| Q925 | 7d | 673.038 | 871.088 | 1311.785 | 836.431 | 1172.687 | 989.917 | 1394.336 | 0.987 | 0.973 | 0.966 | 0.973 | 0.966 | 0.975 | 0.947 |
| | 8d | 773.302 | 999.146 | 1391.247 | 928.037 | 1270.395 | 1067.484 | 1499.221 | 0.983 | 0.964 | 0.958 | 0.965 | 0.960 | 0.972 | 0.938 |
| | 9d | 863.541 | 1113.102 | 1459.822 | 1017.677 | 1351.076 | 1135.576 | 1591.308 | 0.979 | 0.955 | 0.951 | 0.958 | 0.956 | 0.969 | 0.931 |
| | 10d | 943.733 | 1214.456 | 1513.909 | 1093.027 | 1422.259 | 1196.480 | 1666.658 | 0.975 | 0.946 | 0.946 | 0.954 | 0.951 | 0.966 | 0.926 |

