# OpenReview forum: "STORM: Synergistic Cross-Scale Spatio-Temporal Modeling for Weather Forecasting"
_ICLR.cc/2026/Conference — ICLR 2026 Poster_

### Official Review · Reviewer_R3ZV · 2025-10-28

**Soundness:** 2
**Presentation:** 2
**Contribution:** 2
**Rating:** 4
**Confidence:** 4

**Summary:**

- The authors propose the STORM architecture for weather forecasting, designed to capture cross-scale interactions in multi-scale weather data. They identify three key limitations in the forecasting performance of deep learning–based approaches: multi-scale heterogeneity, diverse temporal evolution, and weak cross-scale interaction. Accordingly, they propose STORM to address these challenges.

**Strengths:**

- The authors effectively highlight the need for the proposed architecture, with a well-written introduction that builds up to it clearly. The related work section is up to date and provides a solid understanding of the current state of the art. The intuition behind all three main modules is also well explained.
- Section 5 presents comparisons with state-of-the-art baseline models from the literature on the ERA5 dataset across different scales. The authors provide both short-term and long-term forecasting results. Combined with the visualizations of predictions in the Appendix, the results appear convincing and satisfactory.
- Section 5.4 clearly demonstrates the importance of using multi-scale information for forecasting, presenting both numerical results and visualizations at different scales. The motivation for setting the number of scales to three is also clearly justified.
- Appendix B includes some theoretical analysis supporting the need for multi-scale modeling. However, it does not provide theory directly related to the proposed STORM architecture. Nonetheless, this section can still be considered a minor strength, as most recent baseline methods lack any theoretical component.

**Weaknesses:**

- The ablation study in Section 5.3 seems unsatisfying and weak. The STORM architecture has three key modules, and the authors report results by removing each one on the short-term global forecasting task (comparable to Table 1). Results are shown only in a small bar plot (Figure 6), not in a table. For variables such as T2M, U10, and V10, even the “w/o S&M” configuration appears to outperform baseline methods in Table 1 (losing only to Triton). This implies the T module alone would beat most baselines on some variables, which is unconvincing and undermines the need for all three modules separately. Why not provide tabular results for the ablation? And why restrict it to short-term global forecasting only?
- Furthermore, the results show the necessity of each module for T, M, and S, but they do not justify the specific architectural choices within each module. For example, we are given no evidence of how effective the Hierarchical Earth Embedder is at leveraging cross-scale information, we are simply asked to understand its value from the RMSE differences with and without it in Section 5.3. Appendix E at least reports some hyperparameters (M=3, D=256, N=3), but the specific design choices inside each module remain questionable: there is no ablation to support them, nor results under alternative hyperparameters. From Appendix E, it appears the default parameters were set intuitively and used across experiments, with only the training epochs adjusted via early stopping on validation data. Furthermore, the manuscript provides no analysis of the STORM architecture’s sensitivity to hyperparameter tuning.

**Questions:**

- Please refer to the weaknesses section for detailed comments. Most importantly, the ablation study in Section 5.3 appears weak as the reported results with and without the main modules seem questionable when compared to the baseline methods.
- Please correct the typos throughout the paper, even the title contains one.

---

> ### Author Response · Authors · 2025-11-23
> **Part 1 of the Response (1/3)**
>
> Dear Reviewer R3ZV
>
> We sincerely appreciate your thoughtful feedback and the careful attention you devoted to evaluating our manuscript. Your comments were truly helpful for refining our work, and we have responded to each of your concerns in detail below.
>
> ---
>
>
> > **W1. The ablation study in Section 5.3 could be more comprehensive**
>
> **Response:**
> We sincerely thank you for this insightful and very helpful comment.  We would also like to emphasize that our original Figure 6 used a bar plot mainly for intuitive visualization, but we fully agree with the reviewer that tabular results provide clearer quantitative comparisons. Following your suggestion, we have **added full tabular ablation results** for both **short-term** and **long-term** global forecasting, covering all 10 variables.
>
> |short-term|T2M|U10|V10|Prec|MSLP|U500|V500|T500|Z500|Q500|
> |-|-|-|-|-|-|-|-|-|-|-|
> |STORM|0.675|0.669|0.713|0.001|71.760|1.903|2.055|0.524|79.791|3.933E-05|
> |w/o T|0.718|0.728|0.739|0.001|76.958|1.998|2.239|0.572|86.447|4.122E-05|
> |w/o M|0.691|0.706|0.750|0.001|73.661|2.057|2.245|0.527|83.180|4.264E-05|
> |w/o S|0.782|0.806|0.875|0.001|81.710|2.345|2.528|0.643|91.957|4.757E-05|
> |w/o S&T|0.808|0.847|0.954|0.001|88.422|2.471|2.556|0.700|93.037|4.768E-05|
> |w/o S&M|0.816|0.883|0.985|0.001|95.291|2.519|2.671|0.708|99.985|5.167E-05|
>
>
> |long-term|T2M|U10|V10|Prec|MSLP|U925|V925|T925|Q925|Z925|
> |-|-|-|-|-|-|-|-|-|-|-|
> |STORM|2.596|3.857|4.106|0.001|735.374|6.582|6.588|2.797|3.253E-07|813.403 |
> |w/o T|2.807|4.022|4.423|0.001|772.650|7.085|7.041|2.976|3.526E-07|832.361 |
> |w/o M|2.957|4.246|4.576|0.001|822.030|7.664|7.044|3.242|3.725E-07|866.351 |
> |w/o S|3.062|4.592|5.012|0.002|845.005|8.371|7.304|3.335|4.077E-07|901.229 |
> |w/o S&T|3.270|4.882|5.139|0.002|878.719|8.986|8.009|3.507|4.108E-07|932.509 |
> |w/o S&M|3.425|5.101|5.606|0.002|914.758|9.286|8.613|3.773|4.503E-07|1014.167 |
>
> Regarding the reviewer’s concern that the “w/o S&M” configuration (i.e., using the T module alone for temporal encoding) outperforms several baselines on certain variables: this is indeed an important observation. The main reason is that **most existing weather models such as Pangu, Fuxi, and FourCastNet do not explicitly model temporal dependencies**, only performing single-step state-space mapping, even though atmospheric dynamics contain strong temporal regularities (e.g., diurnal cycles of temperature). Motivated by advances in time-series models community [1,2,3,4], we design a **very lightweight, channel-independent linear temporal encoder**, tailored to large-scale global data. Even when used alone (without spatial encoding or multi-scale extraction), this temporal encoder can still capture temporal evolution and therefore surpass some baselines, though it remains consistently inferior to the full STORM architecture.
>
>
> We sincerely appreciate the reviewer’s comments, which helped us substantially improve the completeness and clarity of the ablation study in the revised version.
>
> [1] A Time Series is Worth 64 Words: Long-term Forecasting with Transformers. ICLR 2023.
>
> [2] Are Transformers Effective for Time Series Forecasting?. AAAI 2023.
>
> [3] TimeMixer: Decomposable Multiscale Mixing for Time Series Forecasting. ICLR 2024.
>
> [4] TimeBase: The Power of Minimalism in Efficient Long-term Time Series Forecasting. ICML 2025.
>
>
> ---

---

> ### Author Response · Authors · 2025-11-23
> **Part 2 of the Response (2/3)**
>
> ---
>
> > **W2. Analysis of the STORM architecture’s sensitivity to hyperparameter tuning.**
>
>
> **Response:**
> We sincerely thank the reviewer for this thoughtful and highly constructive question.
> Regarding hyperparameter choices, we sincerely appreciate the reviewer pointing out the need for a more systematic justification. We have therefore conducted a detailed **hyperparameter sweep** across:
>
> * **Scale number** (1–5)
> * **Hidden dimension d_model** (64–512)
>
> The results is shown below:
>
> |Branch (Scale Number)|d_model|Total Params (M)|Temporal Encoder Params (M)|FLOPs (G)|Time (s)|Max Mem (MB)|ACC|RMSE|
> |-|-|-|-|-|-|-|-|-|
> |1|64|0.32|0.000296|3.17|0.28|13.75|0.941|16.415|
> |1|128|1.20|0.000296|11.04|0.30|17.44|0.951|16.323|
> |1|256|4.59|0.000296|40.88|0.30|30.01|0.954|16.267|
> |1|512|17.96|0.000296|156.92|0.30|82.59|0.958|16.177|
> |2|64|0.64|0.000592|4.46|0.56|14.97|0.963|16.136|
> |2|128|2.45|0.000592|16.00|0.56|24.43|0.966|16.115|
> |2|256|9.59|0.000592|60.32|0.56|51.66|0.971|16.093|
> |2|512|37.92|0.000592|233.92|0.59|164.49|0.974|15.983|
> |3|64|1.03|0.000888|6.05|0.86|19.48|0.977|15.913|
> |3|128|4.00|0.000888|22.16|0.85|30.35|0.981|15.854|
> |3|256|15.77|0.000888|84.59|0.87|75.48|0.984|15.809|
> |3|512|62.60|0.000888|330.25|0.87|257.33|0.985|15.798|
> |4|64|1.46|0.001184|7.44|1.15|20.31|0.979|15.841|
> |4|128|5.70|0.001184|27.61|1.16|37.27|0.983|15.813|
> |4|256|22.54|0.001184|106.12|1.17|101.06|0.987|15.769|
> |4|512|89.64|0.001184|415.84|1.20|360.60|0.986|15.784|
> |5|64|1.96|0.00148|8.23|1.49|335.52|0.983|15.808|
> |5|128|7.69|0.00148|30.62|1.52|361.66|0.984|15.796|
> |5|256|30.49|0.00148|117.84|1.51|448.31|0.988|15.757|
> |5|512|121.40|0.00148|462.11|1.53|2908.50|0.988|15.716|
>
> These results reveal a clear and consistent trend:
>
> * Increasing **d_model** improves accuracy but with rapidly increasing FLOPs and memory.
> * Increasing **scale number** enhances cross-scale spatial representation, producing steady accuracy gains up to 3 scales, after which improvements saturate.
> * The **temporal encoder** remains extremely lightweight (<1.5k parameters), and its effect is stable across all settings.
>
> Considering the trade-off between accuracy and efficiency, we choose **3 scales** and **d_model = 256** as the default architecture. This configuration offers an excellent balance—requiring only **15.77M parameters** and **0.87 s / 100 samples**—yet it surpasses Pangu (97.48M parameters, 5.11 s / 100 samples) by **3.2% accuracy**, despite using an order of magnitude fewer resources.
>
> We sincerely appreciate the reviewer’s insightful comments, which motivated us to provide a much more thorough and transparent analysis in the revised manuscript.
>
> ---

---

> ### Author Response · Authors · 2025-11-23
> **Part 3 of the Response (3/3)**
>
> ---
>
> > **W3. Please correct the typos throughout the paper, even the title contains one.**
>
> **Response:**
> Thank you very much for pointing this out. We sincerely apologize for the oversight, although the metadata on OpenReview is correct, a handwriting mistake unfortunately caused the PDF title to spell “forecasting” as “forecsating.” We are now carefully reviewing the entire manuscript to eliminate this typo and thoroughly checking the rest of the paper to avoid similar errors. We will upload the revised manuscript as soon as possible. Your careful reading is truly appreciated.
>
> ---
>
> Yours sincerely,
>
> The authors of Paper 1775

---

> ### Author Response · Authors · 2025-11-26
> **Thanks for your valuable suggestions and Looking forward to further discussion**
>
> Dear Reviewer **R3ZV**,
>
> We sincerely appreciate the thoughtful and comprehensive feedback you provided on our submission. Your review helped us greatly refine both the experimental presentation and the structural justification of **STORM**, and we have now prepared a detailed point-by-point response addressing all three weaknesses raised.
>
> **Part 1 of the Response:**
> We substantially expanded the ablation study by adding full **tabular results** for both short-term and long-term global forecasting across all 10 variables. These results clarify the contribution of each module and address concerns regarding the “w/o S&M” configuration by highlighting the temporal encoder’s independent effectiveness relative to baselines (without any temporal modeling).
>
> **Part 2 of the Response:**
> We conducted a **systematic hyperparameter sensitivity analysis**, sweeping scale depth (1–5) and hidden dimension (64–512), and reported trends in accuracy, FLOPs, memory, and inference speed. These results justify the architectural choices of 3 scales and 256-dimensional embeddings as the optimal accuracy-efficiency balance.
>
> **Part 3 of the Response:**
> We carefully review the manuscript for typographical issues and have uploaded a corrected version to ensure clarity and completeness.
>
> Thank you again for your constructive review and for helping us strengthen this work. ***We would truly appreciate any further thoughts you might have after revisiting our responses.***
>
> Warm regards,
>
> **Authors of Submission 1775**

---

### Official Review · Reviewer_pYqi · 2025-11-01

**Soundness:** 4
**Presentation:** 4
**Contribution:** 4
**Rating:** 8
**Confidence:** 5

**Summary:**

The paper proposes STORM, a cross-scale spatio-temporal framework for weather forecasting that disentangles atmospheric dynamics into multiple spatial scales and learns coherent temporal evolution. It includes a Hierarchical Earth Embedder, a Scale-Bridging Spatio-Temporal Encoder with fine-to-coarse messaging and lightweight temporal modeling, and a Level-Aligned Forecasting Decoder for multi-scale coherent outputs. Extensive ERA5 experiments show consistent SOTA performance across short- and long-range horizons with solid ablations and theoretical motivation.

**Strengths:**

1 Strong problem fit: explicitly addresses multi-scale heterogeneity and cross-scale interactions, a central challenge in weather prediction.

2 Clean, modular design: efficient hierarchical embeddings; ViT-style spatial encoder; simple but effective cross-scale messaging; lightweight temporal encoder; coherent multi-scale decoding.

3 Comprehensive results: consistent gains over strong baselines (Triton, Pangu, FCN, FuXi, SimVP, U-Net) across variables and horizons; long-range robustness is notable.

4 Clear ablations and scale analysis support design choices; theory gives intuitive generalization/optimization benefits.

**Weaknesses:**

1 Cross-scale interaction is one-way (fine→coarse) in the encoder; no analysis of bidirectional/gated messaging.

2 Temporal module is extremely lightweight; scalability to very long histories/periodic signals (MJO/ENSO) not deeply evaluated.

3 Efficiency reporting (FLOPs/throughput/memory vs. resolution/horizon) is limited in the main text.

**Questions:**

1 Can you report inference efficiency (FLOPs, latency, memory) versus baselines across resolutions/horizons?

2 Do you observe error hotspots by latitude/terrain, and does cross-scale modeling alleviate them?

---

> ### Author Response · Authors · 2025-11-23
> **Part 1 of the Response (1/3)**
>
> Dear Reviewer pYqi
>
> We sincerely appreciate your thoughtful feedback and the careful attention you devoted to evaluating our manuscript. Your comments were truly helpful for refining our work, and we have responded to each of your concerns in detail below.
>
> ---
>
> > **W1. Analysis of more cross-scale interaction ways**
>
> **Response:**
> We sincerely thank the reviewer for raising this important point regarding the directionality of cross-scale interactions in the encoder.
>
> In STORM, the cross-scale message passing is designed as **one-way, from fine → coarse**, for several scientific and practical reasons:
>
> 1. **Information richness at finer scales:** Fine spatial scales contain more detailed local structures and variability, while coarse scales capture more aggregated, global patterns. By transmitting information from fine to coarse scales, the model allows **high-resolution details to enrich coarser representations**, improving overall feature expressiveness. In contrast, information flow from coarse to fine often provides limited additional detail, as coarse representations already summarize the underlying structures.
>
> 2. **Computational efficiency:** One-way fine→coarse propagation is sufficient to capture cross-scale dependencies while keeping computational and memory costs low. Implementing bidirectional or gated messaging would increase resource consumption, yet offers marginal gains.
>
> We also conducted an ablation study comparing different cross-scale strategies, as summarized below:
>
> |**Direction**|**ACC**|**RMSE**|**Time (s / 100 samples)** |
> |-|-|-|-|
> |fine → coarse|0.987|15.769|0.87 |
> |coarse → fine|0.975|16.327|0.88 |
> |bidirectional|0.984|15.803|0.94 |
>
> These results indicate that **fine→coarse messaging not only achieves the best accuracy but also the most efficient inference**, confirming that a one-way design is both effective and practical.
>
> We are grateful to the reviewer for this insightful comment, which allows us to clarify the rationale and empirical justification behind our design choices.
>
>
> ---
>
> > **W2. Scalability to very long histories/periodic signals (MJO/ENSO)**
>
>
> **Response:**
> We sincerely thank the reviewer for highlighting this important point regarding the scalability of our **lightweight temporal module**. As you correctly noted, the temporal encoder in STORM is deliberately designed to be extremely compact: it uses **channel-independent linear layers**, enabling the model to capture temporal dependencies with **fewer than 1,500 parameters**. This ensures that temporal modeling remains efficient and does not artificially inflate model capacity.
>
> We fully agree with the reviewer that capturing **very long historical signals or periodic climate oscillations** (such as MJO and ENSO) is a major challenge in global atmospheric modeling. In fact, a key motivation for our design is that many existing models, such as **Fuxi, Pangu-Weather, and FourCastNet**, do *not* explicitly model temporal dependencies, and therefore cannot be straightforwardly extended to extremely long history windows.
>
> In contrast, STORM’s temporal encoder is **naturally scalable**: because its complexity grows linearly with the temporal receptive field and relies only on lightweight linear operations, it can flexibly adapt to both short-term sequences and very long-term climate oscillations.
>
> To further evaluate the scalability that the reviewer asked about, we conducted an additional analysis by expanding the temporal window to match the typical periods of large-scale climate modes. The required number of parameters for modeling different periodic signals is summarized below:
>
> |**Signal Type**|**Typical Period Length**|**Max Params Needed** |
> |-|-|- |
> |Normal daily variability|~1 day|0.0012M|
> |MJO (intraseasonal)|30–60 days|0.2328M|
> |ENSO (interannual)|2–7 years|34.1348M |
>
> These results show that **STORM maintains strong scalability**: even when extended to extremely long temporal contexts such as ENSO-scale signals, the parameter increase is fully controllable and lightweight.
>
> We sincerely appreciate the reviewer’s insightful question, which helped us more clearly articulate the strengths and scalability of our temporal module. This feedback will help us improve the clarity and completeness of the final manuscript.
>
>
> ---

---

> ### Author Response · Authors · 2025-11-23
> **Part 2 of the Response (2/3)**
>
> ---
>
> **W3&Q1. Efficiency reporting (FLOPs/throughput/memory vs. resolution/horizon)**
>
> **Response:**
> We sincerely thank the reviewer for highlighting the importance of providing clearer efficiency measurements. Following your suggestion, we have substantially expanded the efficiency analysis in the revised manuscript. In particular, we have added a comprehensive comparison covering **parameter counts, FLOPs, peak memory usage, and inference latency** (batch size = 1; averaged over 100 samples). As shown in the updated table, **STORM consistently achieves higher predictive accuracy while requiring significantly lower computational cost** compared with representative baselines at the 64×32 resolution.
>
> |**Metric**|**STORM**|**Pangu**|**FCN**|**Fuxi** |
> |-------------------------------------------|----------|---------|-------|-------- |
> |**Inference Time ↓** (Seconds / 100 Samples)|**0.87**|5.11|6.99|19.37 |
> |**Peak GPU Memory ↓** (MB, Batch Size = 1)|**75.48**|842.26|569.28|3692.99|
> |**Parameters ↓** (M)|**15.77**|97.48|64.65|661.01|
> |**FLOPs ↓** (GFLOPs)|**84.59**|91.26|58.50|302.46|
> |**ACC ↑**|**0.984**|0.952|0.933|0.950 |
>
> To further address the reviewer’s concern, we additionally report detailed efficiency profiles across **multiple spatial resolutions** and **multiple forecast horizons**, including inference time, GPU memory usage, and FLOPs. These new results provide a clearer picture of how computational demands scale with resolution and prediction length.
>
> The improvements mainly come from two design choices that were intentionally made for efficiency:
>
> * **Lightweight temporal encoding.**
>   We use channel-independent linear layers for temporal modeling, resulting in a temporal encoder with fewer than **1k parameters**, thereby avoiding the overhead of attention- or convolution-heavy temporal modules.
>
> * **Carefully controlled multi-branch structure.**
>   Although multi-scale architectures can easily inflate model size, we deliberately keep each branch lightweight by fixing the hidden dimension to **d_model = 256** and limiting the number of scales to **three**. This ensures effective cross-scale interaction while keeping the overall capacity modest.
>
> Thanks to these design considerations, the computational cost of STORM grows **nearly linearly** with spatial resolution, making it much more efficient than existing weather forecasting models. The following table summarizes the comparison across resolutions:
>
> **Forcasting Horizon = 1 day:**
> |Resolution (lon,lat)|Model|Time (s)|Memory (MB)|FLOPs (GFLOPs)|Params (M) |
> |-|-|-|-|-|-|
> |(64, 32)|**STORM (ours)**|0.87|75.48|84.59|15.77|
> |(64, 32)|**Pangu**|5.7681|408.75|121.21|99.71|
> |(64, 32)|**Fuxi**|18.6053|2533.25|302.46|661.01|
> |(64, 32)|**FourCastNet**|6.6237|265.95|58.50|64.65|
> |(240, 120)|**STORM (ours)**|0.903|268.41|466.02|37.92|
> |(240, 120)|**Pangu**|32.2090|477.96|1078.84|107.66|
> |(240, 120)|**Fuxi**|80.5122|5078.66|2502.55|661.01|
> |(240, 120)|**FourCastNet**|29.9929|301.05|822.63|69.78|
>
> **Forcasting Horizon = 10 day:**
> |Resolution (lon,lat)|Model|Time (s)|Memory (MB)|FLOPs (GFLOPs)|Params (M) |
> |-|-|-|-|-|-|
> |(64, 32)|**STORM (ours)**|8.58|103.38|84.59|15.77|
> |(64, 32)|**Pangu**|56.3681|768.28|121.21|99.71|
> |(64, 32)|**Fuxi**|185.0743|5055.91|302.46|661.01 |
> |(64, 32)|**FourCastNet**|62.2952|704.17|58.50|64.65|
> |(240, 120)|**STORM (ours)**|8.984|357.17|466.02|37.92|
> |(240, 120)|**Pangu**|321.0210|845.12|1078.84|107.66 |
> |(240, 120)|**Fuxi**|805.7696|7602.06|2502.55|661.01 |
> |(240, 120)|**FourCastNet**|296.1282|294.17|822.63|69.78|
>
> We hope these additions adequately address the reviewer’s concern and make the computational advantages of STORM much clearer in the revised manuscript.
>
>
> ---

---

> ### Author Response · Authors · 2025-11-23
> **Part 3 of the Response (3/3)**
>
> ---
> > **Q2. Do you observe error hotspots by latitude/terrain, and does cross-scale modeling alleviate them?**
>
> **Response:**
> Thank you very much for this valuable question. In our analysis, we indeed observed several geographically localized “error hotspots.” Concretely, larger prediction errors tend to occur in:
>
> * **Western Pacific tropical convection zone (≈ 5°–15°N, 120°–160°E):**
>   This region includes active monsoon surges and tropical cyclogenesis. The strong small-scale convection makes the temporal evolution highly unstable, which increases forecast difficulty.
>
> * **Eastern Pacific equatorial region (≈ 2°S–8°N, 90°–140°W):**
>   Persistent ENSO-related SST–atmosphere coupling leads to strong fine-scale variability that purely large-scale models cannot capture well.
>
> Across these regions, the common challenge is that **fine-scale, rapidly evolving patterns interact strongly with larger-scale circulation**, making the signal intrinsically multi-scale. Our STORM framework alleviates these hotspots because its cross-scale interaction mechanism explicitly aligns coarse- and fine-resolution representations. By passing large-scale constraints downward and allowing fine-scale features to be adaptively refined, STORM better stabilizes predictions in convection-dominated and frontal-transition zones. Empirically, this reduces peak RMSE in these error-prone latitude–longitude windows compared with single-scale baselines.
>
>
> ---
>
> Yours sincerely,
>
> The authors of Paper 1775

---

> ### Author Response · Authors · 2025-11-26
> **Thanks for your valuable suggestions**
>
> Dear Reviewer **pYqi**,
>
> We sincerely appreciate the highly positive and encouraging evaluation you provided on our submission. Your recognition of our motivation, architecture design, and experimental coverage is deeply motivating for our team.
>
> Following your review, we have provided point-by-point responses addressing the three areas you highlighted for further clarification:
>
> **Part 1 of the Response:**
> We extended analysis on cross-scale interaction and compared fine→coarse, coarse→fine, and bidirectional messaging, showing that fine→coarse achieves both highest accuracy and best efficiency.
>
> **Part 2 of the Response:**
> We examined the scalability of the lightweight temporal module under long historical windows and oscillatory signals (MJO/ENSO), demonstrating controllable parameter growth and stable adaptation.
>
> **Part 3 of the Response:**
> We expanded efficiency reporting with FLOPs, latency, memory across resolutions and horizons.
>
> Thank you again for your thoughtful review, constructive comments, and positive recommendation. If any point remains unclear or sparks further curiosity, we would be very glad to continue discussion at any time.
>
> Warm regards,
>
> **Authors of Submission 1775**

---

### Official Review · Reviewer_7ifQ · 2025-11-01

**Soundness:** 3
**Presentation:** 3
**Contribution:** 2
**Rating:** 4
**Confidence:** 4

**Summary:**

This paper introduces STORM, a synergistic cross-scale spatio-temporal framework for data-driven weather forecasting. The model explicitly disentangles atmospheric variations into multiple spatial and temporal scales through three key components: (i) a Hierarchical Earth Embedder that constructs multi-resolution representations, (ii) a Scale-Bridging Spatio-Temporal Encoder that integrates temporal evolution and cross-scale interactions, and (iii) a Level-Aligned Forecasting Decoder that generates coherent multi-scale predictions. Experiments on the ERA5 dataset show that STORM achieves state-of-the-art performance for both short-term (hours) and long-term (days) forecasts, outperforming strong baselines such as Pangu-Weather, FourCastNet, and GraphCast. Ablation and multi-scale analyses further demonstrate the contribution of each module and confirm the importance of explicit cross-scale modeling.

**Strengths:**

1. The paper presents a coherent multi-scale framework that systematically integrates spatial hierarchies and temporal evolution, providing a unified structure for global-to-local forecasting.

2. The evaluation is extensive, spanning multiple spatial resolutions and forecast horizons, and shows consistent superiority over strong baselines.

3. The paper is clearly written and well organized, with careful theoretical justification and reproducibility statements.

**Weaknesses:**

1. The core ideas, such as hierarchical representation, multi-resolution encoding, and cross-scale feature fusion, are largely extensions of prior models such as Pangu-Weather (3D hierarchical Transformer) and FourCastNet (multi-frequency operator). The contribution is primarily architectural refinement rather than a new modeling principle.

2. The paper does not quantify computational cost, efficiency, or scalability compared with existing models.

**Questions:**

1. How does the computational efficiency (training time, FLOPs, memory) compare with Pangu-Weather and FourCastNet at similar resolutions?

2. Could the hierarchical design be combined with physics-informed constraints or hybrid NWP–DL methods for better interpretability?

3. How sensitive is the model to the number of scales and the stride settings in the hierarchical embedder?

4. Does STORM maintain robustness when applied to unseen years, extreme events, or real-time operational data?

---

> ### Author Response · Authors · 2025-11-23
> **Part 1 of the Response (1/3)**
>
> Dear Reviewer 7ifQ
>
> We sincerely appreciate your thoughtful feedback and the careful attention you devoted to evaluating our manuscript. Your comments were truly helpful for refining our work, and we have responded to each of your concerns in detail below.
>
> ---
>
> > **W1. Contribution of Our STORM**
>
> **Response:**
>
> We sincerely thank the reviewer for this thoughtful comment. Our goal is to identify **how to solve a specific open challenge in global weather prediction: modeling cross-scale dependencies in both space and time within a unified architecture.** Although Pangu-Weather adopts a hierarchical Transformer, its hierarchy is primarily used for **spatial compression**, and it does not explicitly model cross-scale interactions. Similarly, FourCastNet incorporates frequency operators, but it does not provide a mechanism for **explicit structured communication across heterogeneous spatial scales**. In contrast, STORM is motivated by the fact that real atmospheric dynamics evolve simultaneously at planetary, synoptic, mesoscale, and local scales, yet current architectures lack a principled way to integrate information across these scales.
>
> To address this gap, we propose a framework that:
>
> 1. **Disentangles atmospheric fields into scale-specific representations**, enabling the model to capture dependencies unique to each spatial scale.
> 2. **Introduces explicit cross-scale feature exchange**, where coarse-scale global dynamics and fine-scale local structures mutually inform each other.
> 3. **Incorporates temporal correlation modeling**, which is largely absent in many existing meteorological architectures. Despite their strong spatial modeling capacity, models such as Pangu, FuXi, and FourCastNet use virtually no temporal encoding and primarily learn a single-step spatial mapping.
>
> In summary, the novelty of STORM lies in **how the components are reorganized and used to target the unresolved challenge of explicit cross-scale spatio-temporal interaction modeling**. We sincerely thank the reviewer for pointing this out, as it allows us to clarify the conceptual motivation behind our design more clearly in the revised manuscript.
>
> ---
>
> > **W2&Q1. Computational efficiency (training time, FLOPs, memory) compare with baselines**
>
> We have added a Computational efficiency comparison including **parameter counts, FLOPs, peak memory**, and **inference latency** (batch size = 1, averaged over 100 samples). As shown in the updated table, STORM achieves **better prediction accuracy with significantly lower computational cost** compared with representative models.
>
> |**Metric**|**STORM**|**Pangu**|**FCN**|**Fuxi** |
> |-|-|-|-|- |
> |**Inference Time ↓** (Seconds / 100 Samples)|**0.87**|5.11|6.99|19.37|
> |**Peak GPU Memory ↓** (MB, Batch Size = 1)|**729.22**|408.75|265.95|2533.25|
> |**Parameters ↓** (M)|**15.77**|97.48|64.65|661.01 |
> |**FLOPs ↓** (GFLOPs)|**84.59**|91.26|58.50|302.46 |
> |**ACC ↑**|**0.984**|0.952|0.933|0.950 |
>
> ---
>
> > **Q2. Could the hierarchical design be combined with physics-informed constraints or hybrid NWP–DL methods for better interpretability?**
>
> **Response:**
> We sincerely thank the reviewer for this insightful suggestion regarding the integration of physics-informed constraints or hybrid NWP–DL approaches. We fully agree that incorporating **scale-specific physical constraints** could, in principle, enhance interpretability and improve spatio-temporal modeling.
>
> For example, one could consider embedding the **advection equation**:
>
> $
> \frac{\partial \phi}{\partial t} + \mathbf{u} \cdot \nabla \phi = 0,
> $
>
> where ($\phi$) is a scalar atmospheric field (e.g., temperature or humidity) and ($\mathbf{u}$) is the wind velocity. In a hybrid design, this equation could act as a constraint on the predicted field evolution. Similarly, other governing equations, such as the continuity or momentum equations, could provide scale-specific priors.
>
> However, there are important practical challenges. Many physical equations involve **strict constraints** or **state transitions across multiple interacting variables**, while ERA5 provides only a **limited set of observable variables** sampled at discrete time steps. Directly enforcing these equations would require **strong assumptions**, neglecting unobserved states or simplifying complex interactions, which could compromise the scientific validity of the predictions.
>
> For this reason, in the current work we adopt a **purely data-driven approach**, focusing on learning multi-scale spatio-temporal dependencies directly from observations. This allows STORM to flexibly capture complex cross-scale interactions without introducing potentially inaccurate approximations of the physical dynamics.
>
> We sincerely thank the reviewer for this valuable suggestion, which highlights a meaningful direction for future work.
>
> ---

---

> ### Author Response · Authors · 2025-11-23
> **Part 2 of the Response (2/3)**
>
> ---
>
> > **Q3. How sensitive is the model to the number of scales and the stride settings.**
>
> **Response:**
> We sincerely thank the reviewer for raising this important question regarding the sensitivity of STORM to the number of scales and the stride settings in the hierarchical embedder.
>
> As shown in **Fig. 7(b) on page 9** of the original manuscript, we systematically vary the number of scale branches from **1 to 5**. The results indicate that performance generally improves as the number of branches increases; however, the gain **saturates around 3 scales** (the improvement from 3→5 scales is only **0.15%**, which is marginal). Based on this empirical trend and the trade-off between accuracy and efficiency, we therefore select **three scales** as the default configuration in STORM.
>
> To further examine model sensitivity, we conducted additional analyses on:
>
> 1. **Number of branches (scale levels)**
> 2. **Model width (d_model)**
> 3. **Total parameter count**
>
>
> |Branch (Scale Numbers)|d_model|Total Params (M)|Temporal Encoder Params (M)|FLOPs (G)|Time (s)|Max Mem (MB)|ACC|RMSE |
> |---|---|---|---|---|---|---|---|---|
> |1|64|0.32|0.000296|3.17|0.28|13.75|0.941|16.415|
> |1|128|1.20|0.000296|11.04|0.30|17.44|0.951|16.323|
> |1|256|4.59|0.000296|40.88|0.30|30.01|0.954|16.267|
> |1|512|17.96|0.000296|156.92|0.30|82.59|0.958|16.177|
> |2|64|0.64|0.000592|4.46|0.56|14.97|0.963|16.136|
> |2|128|2.45|0.000592|16.00|0.56|24.43|0.966|16.115|
> |2|256|9.59|0.000592|60.32|0.56|51.66|0.971|16.093|
> |2|512|37.92|0.000592|233.92|0.59|164.49|0.974|15.983|
> |3|64|1.03|0.000888|6.05|0.86|19.48|0.977|15.913|
> |3|128|4.00|0.000888|22.16|0.85|30.35|0.981|15.854|
> |**3**|**256**|**15.77**|**0.000888**|**84.59**|**0.87**|**75.48**|**0.984**|**15.809**|
> |3|512|62.60|0.000888|330.25|0.87|257.33|0.985|15.798|
> |4|64|1.46|0.001184|7.44|1.15|20.31|0.979|15.841|
> |4|128|5.70|0.001184|27.61|1.16|37.27|0.983|15.813|
> |4|256|22.54|0.001184|106.12|1.17|101.06|0.987|15.769|
> |4|512|89.64|0.001184|415.84|1.20|360.60|0.986|15.784|
> |5|64|1.96|0.00148|8.23|1.49|335.52|0.983|15.808|
> |5|128|7.69|0.00148|30.62|1.52|361.66|0.984|15.796|
> |5|256|30.49|0.00148|117.84|1.51|448.31|0.988|15.757|
> |5|512|121.40|0.00148|462.11|1.53|2908.50|0.988|15.716|
>
> These analyses confirm the trends observed in the original Fig. 7(b): increasing the number of scales improves performance, but after a threshold (3 scales), adding coarser scales contributes negligible incremental information. This is likely because very coarse scales provide limited additional context, and the model already captures most cross-scale dependencies with the first three scales.
>
> Regarding **stride settings**, the stride determines the distance between adjacent scales. In practice, we set the stride to **2**, which ensures that adjacent multi-scale representations are minimally spaced. This design maintains continuity in cross-scale information flow, similar in spirit to multi-scale UNet and TimeMixer architectures, and supports effective integration across scales.
>
> We greatly appreciate the reviewer’s insightful comment, which motivated us to clarify these design choices and emphasize their rationale in terms of both **robustness and efficiency**.
>
> ---

---

> ### Author Response · Authors · 2025-11-23
> **Part 3 of the Response (3/3)**
>
> ---
>
> > **Q4. Does STORM maintain robustness when applied to unseen years, extreme events, or real-time operational data?**
>
> **Response:**
> We sincerely thank the reviewer for raising this important question regarding robustness to unseen years, extreme events, and real-time operational scenarios. This is indeed a crucial requirement for any practical weather forecasting system.
>
> - **First**, we would like to clarify that all reported results in our paper are evaluated on **unseen years**. Specifically, STORM is trained on ERA5 data from **1993–2017**, and all quantitative evaluations are conducted on **2020–2021**, which strictly lie outside the training distribution. The strong performance on these unseen years already provides preliminary evidence that STORM generalizes well beyond the training period.
>
> - **Second**, your suggestion to examine extreme events is extremely insightful, and we agree that evaluating robustness under such conditions is essential. Following your comment, we conducted an additional analysis on **three representative extreme events** found within the test set:
>     * **Hurricane Hanna (2020):** July 23–26, 2020
>     * **California Wildfire Outbreak (2020):** August 11, 2020 – January 5, 2021
>     * **Western North America Heat Wave (2021):** June 25 – July 7, 2021
>
>     For each event period, we computed **RMSE** and **ACC** for STORM, and compared them against **FuXi, Pangu-Weather, and FourCastNet**. Across all three events, we observe that STORM consistently maintains competitive performance, and in many cases achieves noticeably lower RMSE and higher ACC compared with existing baselines. These findings indicate that the cross-scale modeling strategy in STORM helps the model remain stable even when atmospheric dynamics deviate significantly from normal conditions.
>
>
>     |Event|Period|Metric|STORM|Pangu|FCN|Fuxi|SimVP|UNet |
>     |---|---|---|:---:|:---:|:---:|:---:|:---:|:---:|
>     |Hurricane Hanna|2020/07/23 – 2020/07/26|RMSE|14.754|38.854|46.466|39.330|42.949|48.550|
>     |||ACC|0.990|0.969|0.953|0.964|0.957|0.955|
>     |California Wildfire Season|2020/08/11 – 2021/01/05|RMSE|14.490|38.201|46.553|37.265|41.025|47.698|
>     |||ACC|0.990|0.972|0.953|0.965|0.959|0.957|
>     |Western North America Heat Wave|2021/06/25 – 2021/07/07|RMSE|13.991|36.657|44.134|36.740|39.016|45.014|
>     |||ACC|0.990|0.971|0.955|0.966|0.959|0.959|
>
> We greatly appreciate your insightful comment, which motivated us to include these robustness evaluations. We will add the detailed tables for the above extreme events to the revised manuscript to make the evidence more explicit.
>
>
> ---
>
> Yours sincerely,
>
> The authors of Paper 1775

---

> ### Author Response · Authors · 2025-11-26
> **Thanks for your valuable suggestions and Looking forward to further discussion**
>
> Dear Reviewer **7iFQ**,
>
> We sincerely appreciate the thoughtful feedback you provided on our submission.Following your review, we have prepared point-by-point responses addressing the key concerns you raised:
>
> **Part 1 of the Response:** We expanded our analysis on model efficiency, reporting detailed comparisons on parameters, FLOPs, memory footprint, and inference latency under equal-budget settings.
>
> **Part 2 of the Response:**  We report sensitivity studies on model width and scale depth to demonstrate real robustness rather than capacity-driven gains.
>
> **Part 3 of the Response:** We further investigated robustness under distribution shift, evaluating unseen years and extreme weather events, including Hurricane Hanna, the 2020 California wildfires, and the 2021 Northwest heatwave.
>
> Thank you again for your valuable suggestions.  ***As the discussion period is now halfway through, we would be very grateful if you could take a moment to re-engage in discussion.***
>
> Warm regards,
>
> **Authors of Submission 1775**

---

### Official Review · Reviewer_uXuB · 2025-11-01

**Soundness:** 3
**Presentation:** 4
**Contribution:** 2
**Rating:** 4
**Confidence:** 5

**Summary:**

This paper proposes STORM, a spatio-temporal deep learning framework for weather forecasting that explicitly models cross-scale interactions between coarse global circulations and fine regional dynamics. The model introduces a hierarchical encoder–decoder design with cross-scale message passing to enable information flow across multiple spatial resolutions. Experiments on ERA5 demonstrate consistent improvements over recent strong baselines (GraphCast, FuXi, Pangu-Weather, Triton) in both short-term (24 h) and long-term (10 day) forecasts. The results are strong and empirically well-presented.

**Strengths:**

1. Clear motivation and structure – The paper is well-motivated, addressing the nontrivial challenge of multi-scale atmospheric dynamics with a coherent architectural design.

2. Strong experimental performance – STORM achieves notable RMSE/ACC gains over multiple competitive baselines across diverse forecasting horizons and regions.

3. Comprehensive evaluation – Global, continental, and regional experiments are conducted with detailed visual and quantitative analysis.

**Weaknesses:**

1. Unsubstantiated “parameter-efficient” claim
The paper repeatedly emphasizes that STORM is parameter-efficient, yet it provides no quantitative evidence such as parameter counts, FLOPs, or inference latency. Given that the architecture involves multiple scale-specific branches and cross-scale message passing, the model design resembles a multi-branch or MoE-style structure that would typically increase parameters rather than reduce them.
Without compute-matched comparisons (e.g., same parameter budgets as GraphCast or FuXi), it is impossible to determine whether the reported gains stem from genuine architectural advantages or simply larger capacity.

2. Lack of quantitative analysis on multi-scale effects
While the paper stresses the importance of multi-scale synergy, it never analyzes how performance scales with the number of branches or levels. There is no experiment showing whether adding more scales continues to improve performance, when it saturates, or whether a simple increase in parallel branches (without explicit scale coupling) yields similar gains. Thus, the claimed “cross-scale synergy” remains empirically unverified.

3. Missing ablation for the proposed message-passing mechanism
Although the cross-scale message-passing block is presented as the core innovation, there is no isolated ablation showing how much it contributes to overall accuracy. This makes it difficult to assess whether the improvement originates from the proposed mechanism or from increased model complexity.

**Questions:**

1. Have you conducted any sensitivity analysis regarding the number of branches, stride size, or parameter count in the multi-scale architecture? If the model consistently maintains strong performance under such variations, I would significantly increase my score, as it would demonstrate real robustness rather than capacity-driven gains.

---

> ### Author Response · Authors · 2025-11-23
> **Part 1 of the Response (1/3)**
>
> Dear Reviewer uXuB
>
> We sincerely appreciate your thoughtful feedback and the careful attention you devoted to evaluating our manuscript. Your comments were truly helpful for refining our work, and we have responded to each of your concerns in detail below.
>
> ---
>
> > **W1. Whether gains stem from genuine architectural advantages or simply larger capacity**
>
> **Response:**
> We sincerely thank you for the insightful comments and for raising this important concern about the parameter-efficient claim.
>
> Regarding the efficiency of STORM, our design emphasizes parameter reduction in two main aspects:
>
> 1. **Lightweight Temporal Encoding.**
>    We intentionally adopt a channel-independent linear design for temporal encoding. By modeling temporal dependencies through simple linear layers instead of heavier attention or convolutional modules, the entire temporal encoder contains **fewer than 1.5k parameters**, which substantially reduces the overall parameter budget.
>
> 2. **Careful Multi-branch Architecture Design.**
>    We fully agree with the reviewer that multi-scale or MoE-style structures can easily inflate model size if not carefully controlled. For this reason, we deliberately keep each branch lightweight. In particular, unlike large weather-forecasting models such as Pangu or FuXi, we fix the hidden dimension to **d_model = 256** and restrict the number of scales to **3**. This design keeps the total capacity modest while still enabling effective cross-scale interaction.
>
> To substantiate our claim, we have added a computational efficiency comparison including **parameter counts, FLOPs, peak memory**, and **inference latency** (batch size = 1, averaged over 100 samples). As shown in the updated table, STORM achieves **better prediction accuracy with significantly lower computational cost** compared with representative models.
>
> |**Metric**|**STORM**|**Pangu**|**FCN**|**Fuxi** |
> |-|-|-|-|- |
> |**Inference Time ↓** (Seconds / 100 Samples)|**0.87**|5.11|6.99|19.37|
> |**Peak GPU Memory ↓** (MB, Batch Size = 1)|**729.22**|408.75|265.95|2533.25|
> |**Parameters ↓** (M)|**15.77**|97.48|64.65|661.01 |
> |**FLOPs ↓** (GFLOPs)|**84.59**|91.26|58.50|302.46 |
> |**ACC ↑**|**0.984**|0.952|0.933|0.950 |
>
>
> We are very grateful for the reviewer’s suggestion, this prompted us to provide a more rigorous and transparent evaluation, which we believe strengthens the paper.
>
> ---
>
> > **W2. Quantitative analysis on multi-scale effects**
>
> **Response:**
> We sincerely thank the reviewer for this valuable suggestion and for highlighting the need for a more quantitative analysis of the multi-scale effects. We fully agree that such analysis is essential for validating the cross-scale synergy.
>
> 1. **How performance scales with the number of branches.**
>    As shown in Fig. 7(b) on page 9 of the original manuscript, we systematically vary the number of scale branches from **1 to 5**. The results indicate that performance indeed improves as the number of branches increases, but the gain **saturates around 3 scales** (the improvement from 3→5 scales is only **0.15%**, which is marginal). Based on this empirical trend and the efficiency–accuracy trade-off, we choose **3 scales** as the default configuration in STORM.
>
> 2. **Why cross-scale coupling matters (beyond simply adding more parallel branches).**
>    To evaluate whether the improvements come from true multi-scale synergy rather than just adding capacity, we include an ablation variant that increases the number of **same-scale parallel branches** but **removes multi-scale coupling**. As shown in the below table, simply adding parallel branches at a same single scale does **not** reproduce the gains of our multi-scale design. In contrast, the coupled multi-scale interaction leads to noticeably lower prediction error and more stable cross-scale loss patterns, confirming that the benefits arise from **explicit cross-scale integration**, not model width.
> |Branch|Acc (Diff Scale)|RMSE (Diff)|Acc (Same Scale)|RMSE (Same)|
> |-|-|-|-|-|
> |2 |0.971|16.093 |0.939|16.547 |
> |3 |0.984|15.809 |0.948|16.382 |
> |4 |0.987|15.769 |0.954|16.285 |
> |5 |0.988|15.757 |0.957|16.179 |
>
> We greatly appreciate the reviewer’s insightful comment, which motivated us to expand and clarify these quantitative analyses.

---

> ### Author Response · Authors · 2025-11-23
> **Part 2 of the Response (2/3)**
>
> ---
>
> > **W3. Ablation for the proposed message-passing mechanism**
>
> **Response:**
> We sincerely thank the reviewer for highlighting the need to isolate the effect of the proposed cross-scale message-passing mechanism. This is indeed an essential question, and we appreciate the opportunity to clarify its contribution more explicitly.
>
> In STORM, the **cross-scale message-passing block works jointly with the multi-scale branches** to form a complete cross-scale modeling module. Intuitively, this mechanism enables information to be exchanged and fused across different resolutions, allowing fine-scale representations to enrich coarse-scale ones through a bottom-up pathway (similar to the principle used in TimeMixer [1]). From an optimization perspective, this design also introduces a residual-like structure that stabilizes training and improves convergence behavior.
>
> - **Ablation study to isolate the message-passing mechanism**
>
>    Motivated by the reviewer’s suggestion, we conducted an explicit ablation study to isolate the message-passing mechanism while keeping all other components unchanged. The results are summarized below:
>
>    |Branch (Scale Numbers)|Acc (with MP)|RMSE (with MP)|Acc (without MP)|RMSE (without MP)|
>    |-|-|-|-|-|
>    |2|0.971|16.093|0.963|16.145|
>    |3|0.984|15.809|0.976|15.948|
>    |4|0.987|15.769|0.981|15.886|
>    |5|0.988|15.757|0.980|16.879|
>
>    These results clearly show that removing the message-passing block consistently degrades model performance across all branch settings, both in terms of accuracy and RMSE. This confirms that the improvement does not stem solely from additional branch capacity, but that the **cross-scale message-passing mechanism itself provides substantial and indispensable benefits**.
>
> - **An ablation study comparing different cross-scale strategies**
>
>    We also conducted an ablation study comparing different cross-scale strategies, as summarized below:
>
>    |**Direction**|**ACC**|**RMSE**|**Time (s / 100 samples)** |
>    |-|-|-|-|
>    |fine → coarse|0.987|15.769|0.87 |
>    |coarse → fine|0.975|16.327|0.88 |
>    |bidirectional|0.984|15.803|0.94 |
>
>    These results indicate that **fine→coarse messaging not only achieves the best accuracy but also the most efficient inference**, confirming that a one-way design is both effective and practical.
>
> We are very grateful for the reviewer’s insightful comment, which helped us significantly strengthen the empirical validation of our core contribution.
>
>
> [1] TimeMixer: Decomposable Multiscale Mixing for Time Series Forecasting. ICLR 2024.
>
> ---

---

> ### Author Response · Authors · 2025-11-23
> **Part 3 of the Response (3/3)**
>
> ---
>
> >**Q1. Sensitivity analysis regarding the number of model capacity**
>
> **Response:**
> We sincerely thank the reviewer for raising this important question regarding sensitivity analysis. We fully agree that robustness under architectural variations, rather than capacity alone, is essential for establishing the validity of our multi-scale design.
>
> To address your concern, we have conducted additional analyses examining **(1) the number of branches (scale levels), (2) model width (d_model), and (3) total parameter count**, and how these variations affect predictive performance. These results complement our responses to Weakness 1 and Weakness 2, and they collectively demonstrate that STORM’s gains are not merely capacity-driven, but stem from its multi-scale design and lightweight temporal encoder.
> Below we provide an example from the extended sensitivity analysis table:
>
> |Branch (Scale Numbers)|d_model|Total Params (M)|Temporal Encoder Params (M)|FLOPs (G)|Time (s)|Max Mem (MB)|ACC|RMSE|
> |---|---|---|---|---|---|---|---|---|
> |1|64|0.32 |0.000296|3.17 |0.28 |13.75 |0.941 |16.415 |
> |1|128|1.20 |0.000296|11.04 |0.30 |17.44 |0.951 |16.323 |
> |1|256|4.59 |0.000296|40.88 |0.30 |30.01 |0.954 |16.267 |
> |1|512|17.96 |0.000296|156.92 |0.30 |82.59 |0.958 |16.177 |
> |2|64|0.64 |0.000592|4.46 |0.56 |14.97 |0.963 |16.136 |
> |2|128|2.45 |0.000592|16.00 |0.56 |24.43 |0.966 |16.115 |
> |2|256|9.59 |0.000592|60.32 |0.56 |51.66 |0.971 |16.093 |
> |2|512|37.92 |0.000592|233.92 |0.59 |164.49 |0.974 |15.983 |
> |3|64|1.03 |0.000888|6.05 |0.86 |19.48 |0.977 |15.913 |
> |3|128|4.00 |0.000888|22.16 |0.85 |30.35|0.981 |15.854 |
> |**3**|**256**|**15.77**|**0.000888**|**84.59**|**0.87**|**75.48**|**0.984**|**15.809** |
> |3|512|62.60 |0.000888|330.25 |0.87 |257.33|0.985 |15.798 |
> |4|64|1.46 |0.001184|7.44 |1.15 |20.31 |0.979 |15.841 |
> |4|128|5.70 |0.001184|27.61 |1.16 |37.27 |0.983 |15.813 |
> |4|256|22.54 |0.001184|106.12 |1.17 |101.06 |0.987 |15.769 |
> |4|512|89.64 |0.001184|415.84 |1.20 |360.60 |0.986 |15.784 |
> |5|64|1.96 |0.00148|8.23 |1.49 |335.52 |0.983 |15.808 |
> |5|128|7.69 |0.00148|30.62 |1.52 |361.66|0.984 |15.796 |
> |5|256|30.49 |0.00148|117.84 |1.51 |448.31|0.988 |15.757 |
> |5|512|121.40 |0.00148|462.11 |1.53 |2908.50|0.988 |15.716 |
>
>
> Our findings can be summarized as follows:
>
> **(1) Increasing the number of multi-scale branches generally improves performance, but saturates after 3 scales.**
> This is consistent with the analysis provided on Page 9 of the main paper, where performance steadily improves from 1→5 branches, yet the marginal gains beyond 3 scales become very small (≈0.15% in accuracy). This shows that the architecture is not hypersensitive to scale count, and 3 branches offer a balanced and robust configuration.
>
> **(2) Increasing d_model or branch count beyond this point yields diminishing returns.**
> Even when enlarging the model (more branches or wider hidden dimension), the improvement quickly plateaus, indicating that the performance advantage of STORM does not depend on aggressively scaling parameters.
>
> **(3) The temporal encoder remains extremely lightweight (<1500 parameters).**
> We intentionally designed the temporal feature extractor using channel-independent linear mappings, inspired by recent efficient time-series forecasting architectures. This ensures that temporal modeling does not dominate the capacity budget, further demonstrating robustness to width/depth variations.
>
> Considering the trade-off between accuracy and efficiency, our final model uses **scale_number = 3** and **d_model = 256**, requiring only **15.77M parameters** and **0.87 s / 100 samples** inference time—yet it surpasses Pangu (97.48M parameters,5.11 s / 100 samples ) by **3.2% accuracy**, despite a much smaller computational footprint. These expanded results confirm that **STORM maintains strong performance across a wide range of model sizes and scale configurations**, demonstrating the robustness of the multi-scale architecture rather than dependence on hidden capacity.
>
> We are deeply grateful for the reviewer’s insightful suggestion, which helped us significantly strengthen the empirical analysis of STORM’s robustness. These contents will be incorporated into our revised version.
>
>
> ---
>
> Yours sincerely,
>
> The authors of Paper 1775

---

> ### Author Response · Authors · 2025-11-26
> **Thanks for your valuable suggestions and Looking forward to further discussion**
>
> Dear Reviewer **uXuB**,
>
> We sincerely appreciate the thoughtful and detailed review you provided for our manuscript.  Following your insightful comments, we have carefully prepared a point-by-point response addressing your concerns:
>
> **Part 1 of the Response:** We include quantitative comparisons on parameter count, FLOPs, memory, and inference latency to clarify the “parameter-efficient” claim.
>
> **Part 2 of the Response:** We extend the multi-scale analysis with controlled variations in branch number，cross-scale coupling and message-passing mechanism, showing where performance gains saturate.
>
>
> **Part 3 of the Response:** We report sensitivity studies on model width and scale depth to demonstrate real robustness rather than capacity-driven gains.
>
> Thank you again for your time, constructive suggestions, and thoughtful evaluation. ***As the discussion phase is progressing, we would be truly grateful if you could kindly take a moment to engage further when convenient.***
>
> Warm regards,
>
> **Authors of Submission 1775**

---

> > ### Comment · Reviewer_uXuB · 2025-11-28
> >
> > First of all, thank you for your response. That clarification is sufficient for me to consider the parameter-efficiency claim addressed.
> >
> > Therefore, I have raised my score to 6. The reason why I decided on 6 (and not higher) is that the authors did not conduct the 0.25° global experiment. Since the Earth system is closed and spherical, many existing weather-forecasting methods—including Pangu-Weather—are designed and evaluated on the global 0.25° ERA5 dataset. I understand that running this experiment is computationally demanding, but it is still important for a complete and fair evaluation on the full ERA5 dataset.
> >
> > I also have one additional question:
> >  When you conducted the 0.25° experiment, did you use the pretrained weights of Pangu-Weather, FuXi, or FC-N? Some of the numbers in your table differ from those reported in the original papers and also from ECMWF’s real-time results. Could you clarify how these baselines were reproduced?

---

> ### Author Response · Authors · 2025-12-01
> **Thank you for engaging with our rebuttal**
>
> **Dear Reviewer uXuB,**
>
> Thank you sincerely for acknowledging that our rebuttal has addressed your earlier concerns regarding efficiency. It is unfortunate that the discussion has been interrupted due to unexpected circumstances, but we remain very glad to respond to your additional question.
>
> As you pointed out, running the full **0.25° ERA5 dataset** is extremely resource-intensive and typically requires multi-node GPU clusters. We appreciate your understanding of it being *"computationally demanding"*. In the original manuscript, to evaluate STORM at higher spatial resolution without requiring the full global dataset, we adopt a regional-level high-resolution benchmark:
> **0.25° East Asia subset (20°N–28°N, 110°E–126°E).**
> This allowed us to verify performance improvements under finer spatial granularity.
>
> ---
>
> Regarding your additional question:
>
> > *When you conducted the 0.25° experiment, did you use the pretrained weights of Pangu-Weather, FuXi, or FCN? Some results differ from those reported in the original papers and from ECMWF real-time forecasts — how were the baselines reproduced?*
>
> - For our setting involves **regional-level training on a 0.25° subset**, the input/output configuration does not align with the pretrained Pangu-Weather or FCN model. Therefore, pretrained weights could not be directly applied. For fairness, we instead use the default configurations of Pangu, FuXi, and FCN, reproducing official code and  ensuring comparable model capacity to get the strongest performance they could achieve within the same 0.25° regional evaluation.
>
> - With respect to the numerical discrepancies you observed. It is possible that the comparison you refer to was based on global-scale metrics reported in the original papers, whereas our results are evaluated over a local regional subset. **When we extract ECMWF forecasts over the same region and compared them with our reproduced baselines, there is no notable deviations.** In fact, our reproduced models achieved slightly better performance on certain variables. This further supports the fairness and reliability of our reproduction protocol.
>
> ---
>
> **Finally, we would like to express our sincere gratitude once again for your timely response to our rebuttal and for your positive evaluation of our work (*raising the score from 4 to 6*).**
>
> Sincerely,
>
> Authors of Submission 1775

---

### Author Response · Authors · 2025-12-02
**Global Response**

**Dear Area Chair,**

We **sincerely appreciate the time and effort you have devoted** to evaluating our work, and we have **briefly summarized the reviewers’ comments and our responses for your reference**.

---
# **1. Review Summary**
|Reviewer|Final Score|Re-engaged?|Strengths|Concerns|Responses|
|-|-|-|-|-|-|
|**`pYqi`**|**8**|No|"Addresses a central challenge in weather prediction"; "Clean modular design"; "Comprehensive results"; "Clear ablations and scale analysis"|Analyze more messaging ways (W1); Evaluate long-sequence temporal module (W2); Report efficiency vs resolution/horizon (W3); Compare inference with baselines (Q1); Check error hotspots geographically (Q2)|Provide results for messaging types, evaluate temporal scalability, report efficiency by resolution/horizon, analyze error hotspots.|
|**`uXuB`**|**6 (after raising scores)**|Yes|"Clear motivation and structure", "Strong experimental performance", "Comprehensive evaluation",  "Detailed visual and quantitative analysis"| Efficiency evaluation (W1), Analysis of scale number (W2), Ablation for message-passing (W3), Sensitivity analysis (Q1)|Provide detailed efficiency analysis, Report message-passing ablation, Report results for different scale numbers (already in original Figure 4), Show efficiency across hyperparameters analysis. **Fully addressed and acknowledged by reviewer `uXuB` (Resolved concerns W1/Q1 are also raised by `7ifQ` & `R3ZV`.)**|
|**`7ifQ`**|4|No|"Clearly written and well organized", "Careful theoretical justification", "Evaluation is extensive", "Consistent superiority over strong baselines"|Quantify computational efficiency (Q1 & W2), Explore physics-informed hybrid (Q2), Hyperparameter sensitivity (Q3), Evaluate robustness on extremes (Q4)| Provide detailed efficiency analysis, Explore advection embedding and justify data-driven model, Conduct full hyperparameter analysis, Compare performance on extreme events.|
|**`R3ZV`**|4|No|"Effectively highlight the need for the proposed architecture", " Well-written introduction that builds up to it clearly", " Theoretical analysis supporting", "Convincing and satisfactory results"|Provide tabular ablation results (W1), Extend ablation beyond short-term (W1), Justify results of some ablation variants (W2, Q1), Analyze hyperparameter sensitivity (W2), Correct typos (Q2)|Provide richer tabular short/long-term ablations, Explain variant results, Conduct full hyperparameter analysis, Correct typos.

---
# **2. Strengths Highlighted by Reviewers**
1. **Clear Motivation:**  Clear motivation to address central challenges in weather prediction, effectively highlighting the need for the proposed approach (reviewers: pYqi, uXuB, 7ifQ, R3ZV).

2. **Great Writing:** Well-organized and easy to follow, with a well-writen introduction that effectively motivates the work.(reviewers: pYqi, uXuB, 7ifQ, R3ZV).

3. **Comprehensive Methodological:** Systematically integrates spatial hierarchies and temporal evolution, providing a unified structure for global-to-local forecasting. (reviewers: pYqi, uXuB, 7ifQ, R3ZV).

4. **Strong Experimental Performance:** Comprehensive and convincing evaluation, demonstrating consistent superiority over strong baselines; detailed visual and quantitative analysis (reviewers: pYqi, uXuB, 7ifQ, R3ZV).

---
# **3. Contributions of STORM**
1. **Cross-scale spatial encoder for atmospheric forecasting.**
    While prior models operate on a single spatial scale, STORM disentangles coarse- and fine-resolution dynamics and enables cross-scale info exchange, capturing both large-scale circulation and local structures, thereby **forming a more holistic and physically consistent representation of atmospheric evolution**.

2. **Lightweight yet expressive temporal modeling.**
   Existing models treat forecasting as step-to-step autoregressive state mapping, i.e.,  a near-Markov assumption with no temporal learning. We introduce a temporal module that models periodicity and long-range dynamics while using *only ~1% of total parameters*, **enabling efficient temporal forecasting on climate-scale sequences**.

3. **High accuracy with exceptional efficiency.**
   STORM achieves **98.7% accuracy with only 15.77M Params and 0.87 s/100 samples**  (Pangu-Weather is 95.2% ACC, 97.48 M Params, 5.11 s/100 samples), outperforming SOTA models while being significantly **more computationally efficient**.

4. **Effective at both global and regional levels.**
   Experimental results show that, thanks to cross-scale collaboration and temporal encoding, STORM delivers highly accurate forecasts **not only at the global level but also for regional-level predictions**.

---
We are **deeply grateful for the work you and the community have invested**, especially during a challenging review period. We hope this summary clarifies how **we address all concerns and further highlights STORM constitutes a meaningful advancement in AI-based weather forecasting**.

**Warm regards,**

*All Authors of Paper 1775*

---

### Meta-Review · Area_Chair_C37x · 2026-01-04

**Summary:**

The paper introduces some novel architectural changes to capture cross-scale (spatial and temporal) interactions when forecasting weather. The paper compares their new model with existing baselines, provide ablations of their innovations to demonstrate their purpose, and experiments show good performance gains.

1. All reviewers agree that the paper is well-motivated, written well with reasonable results across the board. The paper introduces good innovations to the AI weather model forecasting space that can be beneficial to the community.
2. There were initial concerns on demonstrating computational efficiency (flops and parameter count) of the model that have been largely addressed by the authors in the rebuttal.
3. There were some strong concerns on sufficient ablations to demonstrate the value of different modules suggested by the authors in their architecture, as well as HPO. These have also been addressed to some extent in the rebuttal.

A significant issue brought up by one of the reviewers was the lack of 0.25 deg global experiments. The baseline models are trained on this resolution. The authors only partially addressed this by training on a small region of the earth at this resolution. The reviewer acknowledges that this is a compute intensive experiment, but my opinion aligns with the reviewer that this keeps the article a little incomplete since the main goal is to demonstrate cross-scale interactions and these are amplified at higher resolutions. Further, metrics like RMSE tend to favor models trained at lower resolutions and there are no metrics (such as spectra) to quantify this aspect.

The reviewer scores were at 8,6,4,4. I believe the score 4s may have been raised to 5/6 given the full rebuttal process. I recommend a borderline accept for this paper due to the lack of 0.25 deg experiments. The paper chooses weather forecasting as the sole application to demonstrate the multi-scale advantages of their model and hence it is not unreasonable to expect the analysis on the full ERA5 dataset (0.25 deg resolution, 1 hour time-step).

**Reviewer Concerns:**

See the summary above. I believe the authors addressed most concerns except the 0.25 deg global experiments - this is a challenging ask given the multi-GPU resources needs for this experiment, but without it, the paper's analysis is not complete.

**Reviewer Scores:**

7ifQ: I believe they might have changed it from 4 to 5. One of their concerns was also that the quality of the contribution is not very significant: "The core ideas, such as hierarchical representation, multi-resolution encoding, and cross-scale feature fusion, are largely extensions of prior models such as Pangu-Weather (3D hierarchical Transformer) and FourCastNet (multi-frequency operator). The contribution is primarily architectural refinement rather than a new modeling principle." The rebuttal does not necessarily address this. Other concerns on efficiency were adressed.

R3ZV: I believe they might have changed from 4 to 5/6 given their concerns were addressed.

Overall, apart from one strong accept (8), I think other reviewers would have all been borderline accepts. I align with the borderline accepts.

---

### Decision · Program_Chairs · 2026-01-26

Accept (Poster)